# Alternating Mirror Descent
# for Constrained Min-Max Games

**Andre Wibisono**
Department of Computer Science
Yale University
andre.wibisono@yale.edu

**Molei Tao**
Department of Mathematics
Georgia Institute of Technology
mtao@gatech.edu

**Georgios Piliouras**
Engineering Systems and Design
Singapore University of Technology and Design
georgios@sutd.edu.sg

## Abstract

In this paper we study two-player bilinear zero-sum games with constrained strategy spaces. An instance of natural occurrences of such constraints is when mixed strategies are used, which correspond to a probability simplex constraint. We propose and analyze the alternating mirror descent algorithm, in which each player takes turns to take action following the mirror descent algorithm for constrained optimization. We interpret alternating mirror descent as an alternating discretization of a skew-gradient flow in the dual space, and use tools from convex optimization and modified energy function to establish an $O(K^{-2/3})$ bound on its average regret after $K$ iterations. This quantitatively verifies the algorithm's better behavior than the simultaneous version of mirror descent algorithm, which is known to diverge and yields an $O(K^{-1/2})$ average regret bound. In the special case of an unconstrained setting, our results recover the behavior of alternating gradient descent algorithm for zero-sum games which was studied in [2].

## 1 Introduction

Multi-agent systems are ubiquitous in many applications and have gained increasing importance in practice, from the classical problems in economics and game theory, to the modern applications in machine learning, in particular via the emergence of learning strategies that can be formulated as min-max games, such as the generative adversarial networks (GANs), robust optimization, reinforcement learning, and many others [11, 16, 23, 27]. In multi-agent systems, interaction between the agents can exhibit nontrivial global behavior, even when each agent individually follows a simple action such as a greedy, optimization-driven strategy [7, 8, 1]. This emergence of complex behavior has been recognized as one source of difficulties in understanding and controlling the global behavior of multi-agent game dynamics [24, 15].

Even in the basic setting of unconstrained two-player zero-sum game with bilinear payoffs, this emergence of non-trivial behavior already presents some difficulties. It is now well-known that if each player follows a classical greedy strategy, such as gradient descent, and if they make their actions *simultaneously*, then their joint trajectories diverge away from equilibrium and lead to increasing regret; however, the average iterates still converge to the equilibrium and yield a vanishing average regret with decreasing step size [6, 3, 7]. This behavior challenges our intuition and is in marked contrast to the case of single-agent optimization, in which greedy strategies are guaranteed to converge. This leads to variations of the greedy strategy to correct the diverging behavior and help

36th Conference on Neural Information Processing Systems (NeurIPS 2022).

the trajectories to converge to equilibrium, for example via optimistic or extragradient versions of gradient descent [10, 17], which can be seen as approximations of the proximal (implicit) gradient descent [19].

Another variation of the basic greedy strategy is when each player follows gradient descent, but they make their actions in an *alternating* fashion (i.e. one at a time, rather than simultaneously), as studied in [2]. This technique is particularly useful for machine learning applications where the state of the system can be very large with several billions of parameters as one does not need extra memory to store intermediate variables, required both by simultaneous updates as well as extra-gradient methods. The resulting *alternating gradient descent* algorithm has a markedly different behavior than in the simultaneous case. As shown in [2], the trajectories of alternating gradient descent turn out to *cycle* (stay in a bounded orbit), rather than converging to or diverging away from equilibrium. This behavior mimics the ideal setting of continuous-time dynamics, which is the limit as the step size goes to $0$, in which case the orbit of the two players cycles around the equilibrium and exactly preserves the "energy function", which in this case is defined to be the distance to the equilibrium point, achieving average regret bound $O(T^{-1})$ after (continuous) time $T$ [18]. In discrete time, alternating gradient descent does not exactly conserve the energy function; instead, it conserves a *modified energy function*, which is a perturbation of the true energy function with a correction term which is proportional to the step size. This implies that alternating gradient descent has a constant regret for any step size, and thus yields an average regret bound of $O(K^{-1})$ after $K$ iterations, which matches the continuous-time behavior; see [2] for more details.

The results above raise the question of what we can say in the *constrained* setting, when each agent can choose an action from a constrained set. An instance of natural occurrences of such constraints is when mixed strategies are used, which corresponds to a probability simplex constraint. One popular strategy, inspired by optimization techniques, is that each agent now plays the *mirror descent* algorithm for constrained minimization of their own objective function. Mirror descent is closely related to the Follow the Regularized Leader (FTRL) algorithm in online learning, which has a Hamiltonian structure in continuous time [4, 13]. In the idealized continuous-time setting, if both players follow the continuous-time version of mirror descent dynamics, then their trajectories cycle around the equilibrium, and conserve an "energy function" in the dual space, which is now defined to be the sum of the dual functions of the regularizers; this leads to a constant regret, and yields an $O(T^{-1})$ average regret bound after (continuous) time $T$ [18]. In this harder constrained setting, one would hope to weave a single thread connecting in an intuitive manner the behavior of continuous dynamics with multiple distinct discretizations. In discrete time, if the two players follow the simultaneous version of mirror descent, then their trajectories diverge from equilibrium, and yields a $O(K^{-1/2})$ bound on the average regret. If both players follow the *proximal* (implicit) mirror descent algorithm, then as we show their trajectories converge to the equilibrium, resulting in a simple a $O(K^{-1})$ average regret bound, which matches the continuous-time behavior. While the $O(K^{-1})$ average regret bound is known to extend to general games [25] under clairvoyant multiplicative weights updates, which is a closely related algorithm to proximal mirror descent, such methods require coordination among the players to solve the implicit update. Nevertheless, there are simpler variations such as the optimistic or extra-gradient, which can get $\tilde{O}(K^{-1})$ convergence rates in general games [9], where $\tilde{O}$ hides polylogarithmic dependence.

In this paper, we propose and study the *alternating mirror descent* algorithm for two-player zero-sum constrained bilinear game, in which the two players take turns to make their moves following the mirror descent algorithm. We show that the total regret of the two players can be expressed in terms of a *modified energy function*, which generalizes the modified energy function in the unconstrained setting (see Theorem 4.5). Recall in the unconstrained setting, the modified energy function is exactly preserved [2]. In the constrained setting, we prove a bound on the growth of the modified energy under third-order smoothness assumption on the energy function (see Theorem 4.4). This yields an $O(K^{-2/3})$ bound on the average regret of the alternating mirror descent algorithm, which improves on the classical $O(K^{-1/2})$ average regret bound of the simultaneous mirror descent algorithm.

As an analysis tool, we study alternating mirror descent algorithm as an alternating discretization of a *skew-gradient flow*. In continuous time, we can study the continuous-time mirror descent dynamics in the dual space; this dynamics corresponds to the skew-gradient flow of the energy function, which conserves the energy and explains the cycling behavior, as we explain in Section 4. In discrete time, since the energy function is convex, the forward discretization of the skew-gradient

flow is proved to increase energy; this corresponds to the diverging behavior of the simultaneous mirror descent algorithm; see Section B.1. In contrast, the backward (implicit) discretization of the skew-gradient flow is proved to decrease energy; this corresponds to the converging behavior of the proximal/clairvoyant learning; see Section B.2. Finally, as in the unconstrained case, the alternating discretization of the flow follows the continuous-time dynamics more closely, leading to improved bounds.

Another reason that we expect the alternating mirror descent algorithm to work well is a connection to symplectic integrators. In the special case when the payoff matrix in the bilinear game is the identity matrix,[1] the continuous-time mirror descent dynamics becomes a *Hamiltonian flow*, i.e. it has a symplectic structure, and the energy function becomes the Hamiltonian function that generates the flow (hence conserved). In discrete time, the alternating mirror descent algorithm corresponds to the *symplectic Euler* discretization in the dual space, which also conserves the symplectic structure. A remarkable feature of symplectic integrators is that they exhibit good energy conservation property until exponentially long time [5, 12]. Recently, symplectic integrators as well as connections between algorithms and continuous-time dynamics, have been shown to be highly relevant in the optimization settings for design of accelerated methods [14, 30, 28, 20]. Specifically, by preserving specific continuous symmetries of the flow, symplectic integrators stabilize the dynamics and allow for larger step sizes for fixed accuracy in the long run. This intuition gives a geometric interpretation to "acceleration". Our novel connection between game theory, alternating mirror descent dynamics and symplectic integrators opens the door for further cross-fertilization between these areas.

## 2 Preliminaries

In this paper we study the two-player zero-sum game with bilinear payoff:

$$\min_{p \in \mathcal{P}} \max_{q \in \mathcal{Q}} p^\top A q.$$

Here $\mathcal{P} \subseteq \mathbb{R}^m$ and $\mathcal{Q} \subseteq \mathbb{R}^n$ are closed convex sets which represent the domains of the actions that each player can make, and $A \in \mathbb{R}^{m \times n}$ is an arbitrary payoff matrix. When the first player plays an action $p \in \mathcal{P}$ and the second player plays $q \in \mathcal{Q}$, the first player receives loss $p^\top A q = \sum_{i=1}^{m} \sum_{j=1}^{n} p_i q_j A_{ij}$, which they want to minimize; while the second player receives loss $-p^\top A q$, which they want to minimize (equivalently, the second player wants to maximize their utility $p^\top A q$).

An objective of the game is to reach the *Nash equilibrium*, which is a pair $(p^*, q^*) \in \mathcal{P} \times \mathcal{Q}$ that satisfies, for all $(p, q) \in \mathcal{P} \times \mathcal{Q}$:

$$(p^*)^\top A q \leq (p^*)^\top A q^* \leq p^\top A q^*.$$

By von Neumann's min-max theorem [29], a Nash equilibrium always exists (but is not necessarily unique) as long as $\mathcal{P}$ and $\mathcal{Q}$ are compact, which is the case here.

One way to measure convergence to equilibrium is via the *duality gap* $\mathsf{dg} : \mathcal{P} \times \mathcal{Q} \to \mathbb{R}$ given by

$$\mathsf{dg}(p, q) = \max_{\tilde{q} \in \mathcal{Q}} p^\top A \tilde{q} - \min_{\tilde{p} \in \mathcal{P}} \tilde{p}^\top A q.$$

One can check that $\mathsf{dg}(p, q) \geq 0$ for all $(p, q) \in \mathcal{P} \times \mathcal{Q}$, and moreover, $\mathsf{dg}(p^*, q^*) = 0$ if and only if $(p^*, q^*)$ is a Nash equilibrium.

In the execution of the zero-sum game, each player follows some algorithm in discrete time (or some dynamics in continuous time). Depending on the precise specification of what algorithms they play and how they play it, the iterates of the actions $(p_k, q_k)$ may not converge to the Nash equilibrium. However, we expect the *average iterates* $(\bar{p}_K, \bar{q}_K)$ to converge to the Nash equilibrium, as measured by the duality gap: $\mathsf{dg}(\bar{p}_K, \bar{q}_K) \to 0$ as $K \to \infty$. Furthermore, typically the duality gap of the average iterates are related to the *average regret* of the two players. Therefore, if both players follow a *no-regret* algorithm (so the average regret vanishes asymptotically), then their average iterates converge to Nash equilibrium. Thus, we are interested in bounding the rate at which the average regret of the two players converges to 0. See Section 3.2 for more details.

---

[1]When the payoff matrix is arbitrary, the dynamics has a non-canonical Hamiltonian structure.

**Unconstrained case.** A special case is the *unconstrained* setting when $\mathcal{P} = \mathbb{R}^m$ and $\mathcal{Q} = \mathbb{R}^n$. Here the equilibrium is at $(0,0)$. A natural strategy is for each player to follow gradient descent to minimize their own loss functions. As explained in the introduction, the behaviors of the iterates can vary depending on how the two players take the actions: If they both follow simultaneous gradient descent, then the iterates diverge; if they follow simultaneous proximal gradient descent, then the iterates converge; if they follow continuous-time gradient flow, then the iterates cycle; finally, if they follow alternating gradient descent, then the iterates cycle and conserve a modified energy function.

**Constrained case.** The main question we address in this paper is whether the different behaviors we observe in the unconstrained setting above carry over to the *constrained* setting, when $\mathcal{P} \subsetneq \mathbb{R}^m$ or $\mathcal{Q} \subsetneq \mathbb{R}^n$ (or both) are proper subsets of the Euclidean space. As an example, we may consider $\mathcal{P} = \Delta_m$ and $\mathcal{Q} = \Delta_n$, where $\Delta_m = \{x \in \mathbb{R}^m \colon x_i \geq 0, \sum_{i=1}^m x_i = 1\}$ is the probability simplex in $\mathbb{R}^m$, and similarly $\Delta_n$ is the probability simplex in $\mathbb{R}^n$; these correspond to the setting where each player chooses a random action among a set of discrete choices. However, our analyses and results hold for general constraint sets.

In this constrained setting, a natural strategy is for each player to follow a constrained greedy method, instead of using gradient descent. Following techniques in optimization, we consider the case when each player follows the *mirror descent* algorithm [21] to minimize their own loss functions over their constrained domains.

**Mirror descent set-up.** We assume that on the domain $\mathcal{P} \subseteq \mathbb{R}^m$ we have a strictly convex regularizer function $\phi \colon \mathcal{P} \to \mathbb{R}$ which is a *Legendre function* [26], which means $\phi$ is continuously differentiable, $\|\nabla \phi(p)\| \to \infty$ as $p$ approaches the boundary $\partial \mathcal{P}$ of $\mathcal{P}$, and $\nabla \phi$ is a bijection from $\mathcal{P}$ to the range $\nabla \phi(\mathcal{P}) \subseteq \mathbb{R}^m$. Here the gradient $\nabla \phi(p) \in \mathbb{R}^m$ is the vector of partial derivatives.

Let $D_\phi \colon \mathcal{P} \times \mathcal{P} \to \mathbb{R}$ be the *Bregman divergence* of $\phi$, defined by

$$D_\phi(p, \tilde{p}) = \phi(p) - \phi(\tilde{p}) - \langle \nabla \phi(\tilde{p}), p - \tilde{p} \rangle$$

where $\langle u, v \rangle = u^\top v$ is the $\ell_2$-inner product. Since $\phi$ is strictly convex, we have that $D_\phi(p, \tilde{p}) \geq 0$, and $D_\phi(p, \tilde{p}) = 0$ if and only if $p = \tilde{p}$. Note that Bregman divergence is not necessarily symmetric: in general, $D_\phi(p, \tilde{p}) \neq D_\phi(\tilde{p}, p)$. The idea of mirror descent [21] is to use the Bregman divergence $D_\phi(p, \tilde{p})$ to measure "distance" (albeit asymmetric) from $\tilde{p}$ to $p$ on $\mathcal{P}$. For example, in the unconstrained setting when $\mathcal{P} = \mathbb{R}^m$, if we choose $\phi(p) = \frac{1}{2}\|p\|_2^2$ where $\|p\|_2^2 = p^\top p$ is the (squared) $\ell_2$-norm, then the Bregman divergence recovers the standard $\ell_2$-distance: $D_\phi(p, \tilde{p}) = \frac{1}{2}\|p - \tilde{p}\|_2^2$, which is symmetric. On the other hand, if $\mathcal{P} = \Delta_m \subset \mathbb{R}^m$ and we choose $\phi(p) = \sum_{i=1}^m p_i \log p_i$ to be negative entropy, then the Bregman divergence is the relative entropy or the Kullback-Leibler divergence: $D_\phi(p, \tilde{p}) = \sum_{i=1}^m p_i \log \frac{p_i}{\tilde{p}_i}$, which is not symmetric.

Similarly, we assume that on the domain $\mathcal{Q} \subseteq \mathbb{R}^n$ we have a strictly convex regularizer function $\psi \colon \mathcal{Q} \to \mathbb{R}$ which is a Legendre function, so $\psi$ is continuously differentiable, $\|\nabla \psi(q)\| \to \infty$ as $q \to \partial \mathcal{Q}$, and $\nabla \psi$ is a bijection from $\mathcal{Q}$ to the range $\nabla \psi(\mathcal{Q}) \subseteq \mathbb{R}^n$. Let $D_\psi \colon \mathcal{Q} \times \mathcal{Q} \to \mathbb{R}$ be the Bregman divergence of $\psi$, defined by $D_\psi(q, \tilde{q}) = \psi(q) - \psi(\tilde{q}) - \langle \nabla \psi(\tilde{q}), q - \tilde{q} \rangle$.

## 3 Algorithm: Alternating Mirror Descent

We consider the **Alternating Mirror Descent (AMD)** algorithm where each player follows the mirror descent algorithm to minimize their own loss function, and they perform the updates in an *alternating* fashion (one at a time).

Concretely, the AMD algorithm starts from an arbitrary initial position $(p_0, q_0) \in \mathcal{P} \times \mathcal{Q}$. At each iteration $k \geq 0$, suppose the players are at position $(p_k, q_k) \in \mathcal{P} \times \mathcal{Q}$. Then in the next iteration $k + 1$, they update their position to:

$$p_{k+1} = \arg\min_{p \in \mathcal{P}} \left\{ p^\top A q_k + \tfrac{1}{\eta} D_\phi(p, p_k) \right\} \tag{1a}$$

$$q_{k+1} = \arg\min_{q \in \mathcal{Q}} \left\{ -p_{k+1}^\top A q + \tfrac{1}{\eta} D_\psi(q, q_k) \right\} \tag{1b}$$

where $\eta > 0$ is step size. Observe the first player (the $p$ player) makes the update first using the current value $q_k$; then the second player (the $q$ player) updates using the new value $p_{k+1}$. We assume both players can solve the update equations (1), which is the case because they can choose the convex regularizers $\phi, \psi$. Observe that we can write the optimality condition for AMD (1) as:

$$\nabla\phi(p_{k+1}) = \nabla\phi(p_k) - \eta A q_k \tag{2a}$$

$$\nabla\psi(q_{k+1}) = \nabla\psi(q_k) + \eta A^\top p_{k+1}. \tag{2b}$$

In Section 4.1, we interpret AMD as an alternating discretization of a skew-gradient flow in dual space.

## 3.1 Continuous-Time Motivation: Skew-Gradient Flow

One way to derive the alternating mirror descent algorithm (1) is as a discretization of what we call the *skew-gradient flow* in continuous time. Concretely, let us define the dual functions (convex conjugate) of the convex regularizers $\phi^*\colon \mathbb{R}^m \to \mathbb{R}$ and $\psi^*\colon \mathbb{R}^n \to \mathbb{R}$ given by:

$$\phi^*(x) = \sup_{p\in\mathcal{P}} \ \langle p, x \rangle - \phi(p)$$

$$\psi^*(y) = \sup_{q\in\mathcal{Q}} \ \langle q, y \rangle - \psi(q).$$

We call $\mathbb{R}^m$ and $\mathbb{R}^n$ (the domains of $\phi^*$ and $\psi^*$) as the *dual space*. Recall that the gradient of the dual function is the inverse map of the original gradient: $\nabla\phi^* = (\nabla\phi)^{-1}$ and $\nabla\psi^* = (\nabla\psi)^{-1}$.

Given $(p_k, q_k) \in \mathcal{P} \times \mathcal{Q}$, we define the **dual variables** $(x_k, y_k) \in \mathbb{R}^{m+n}$ by:

$$x_k = \nabla\phi(p_k) \qquad \Leftrightarrow \qquad p_k = \nabla\phi^*(x_k)$$

$$y_k = \nabla\psi(q_k) \qquad \Leftrightarrow \qquad q_k = \nabla\psi^*(y_k).$$

Suppose $(p_k, q_k) \in \mathcal{P} \times \mathcal{Q}$ evolves following AMD (1), so they satisfy the optimality condition (2). Then the dual variables $(x_k, y_k) \in \mathbb{R}^{m+n}$ evolve by the following algorithm:

$$x_{k+1} = x_k - \eta A \nabla\psi^*(y_k) \tag{3a}$$

$$y_{k+1} = y_k + \eta A^\top \nabla\phi^*(x_{k+1}). \tag{3b}$$

We refer to AMD update in the dual space (3) above as the *alternating method*.

**Continuous-time limit.** Observe that as the step size $\eta \to 0$, the update equation (3) for AMD in the dual space recovers the following continuous-time dynamics for $(X_t, Y_t) \in \mathbb{R}^{m+n}$:

$$\dot{X}_t = -A\nabla\psi^*(Y_t) \tag{4a}$$

$$\dot{Y}_t = A^\top\nabla\phi^*(X_t). \tag{4b}$$

Here $\dot{X}_t = \frac{d}{dt}X_t$ is the time derivative. We call the dynamics (4) as the *skew-gradient flow*, generated by the *energy function*:

$$H(x, y) := \phi^*(x) + \psi^*(y). \tag{5}$$

A particular feature of the skew-gradient flow (4) is that it preserves the energy function over time:

$$H(X_t, Y_t) = H(X_0, Y_0)$$

for all $t \geq 0$; see Section 4 for further detail.

**AMD in dual space as alternating discretization of skew-gradient flow.** We can view the update equation (2) as performing an alternating discretization of the skew-gradient flow (4). Since the energy function is separable, these updates are explicit, and correspond to performing alternating mirror descent in the dual space (3). See Section 4.1 for more detail.

## 3.2 Regret Analysis of Alternating Mirror Descent

To measure the performance of the algorithm, we can analyze the *regret* of each player, which is the gap between their observed losses and the best loss they could have achieved in hindsight, using a fixed (static) action. We define the regret of the alternating mirror descent algorithm (1) as follows.

**Regret definition.**    From iteration $k$ to iteration $k+1$ of the algorithm, there are two half steps that happen: The first player updates from $p_k$ to $p_{k+1}$ (while the second player is at $q_k$), then the second player updates from $q_k$ to $q_{k+1}$ (while the first player is at $p_{k+1}$). Thus, the first player observes $q_k$ twice: once when the first player is at $p_k$, and once when they are at $p_{k+1}$. Therefore, we define the regret of the first player after $K$ iterations, with respect to a static action $p \in \mathcal{P}$, to be:

$$R_{1,K}(p) := \sum_{k=0}^{K-1} \left( \frac{p_k + p_{k+1}}{2} \right)^\top A q_k - \sum_{k=0}^{K-1} p^\top A q_k. \tag{6}$$

Similarly, the second player observes $p_{k+1}$ twice: once when the second player is at $q_k$, and once when they are at $q_{k+1}$. Therefore, we define the regret of the second player after $K$ iterations, with respect to a static action $q \in \mathcal{Q}$, to be:

$$R_{2,K}(q) := \sum_{k=0}^{K-1} p_{k+1}^\top A q - \sum_{k=0}^{K-1} p_{k+1}^\top A \left( \frac{q_k + q_{k+1}}{2} \right). \tag{7}$$

Note that in the unconstrained case, this recovers the regret definition of the alternating gradient descent algorithm of [2] (up to a factor of $\frac{1}{2}$ for normalization).

We define the cumulative regret of both players after $K$ iterations, with respect to static actions $(p, q) \in \mathcal{P} \times \mathcal{Q}$, to be:

$$R_K(p, q) := R_{1,K}(p) + R_{2,K}(q).$$

Finally, we define the **total regret** of both players after $K$ iterations to be the best cumulative regret in hindsight:

$$R_K := \sup_{(p,q) \in \mathcal{P} \times \mathcal{Q}} R_K(p, q).$$

**Regret and duality gap.**    We define the average iterates of the players after $K$ iterations to be:

$$\bar{p}_K = \frac{1}{K} \sum_{k=0}^{K-1} p_{k+1} \qquad \text{and} \qquad \bar{q}_K = \frac{1}{K} \sum_{k=0}^{K-1} q_k.$$

Note that we shift the index of the first player by one, since the second player moves after the first player. Then we observe that the total regret is related to the duality gap of the average iterates. We provide the proof of Lemma 3.1 in Section C.1.1.

**Lemma 3.1.** *Under the above definitions, for any $K \geq 1$,*

$$\mathsf{dg}(\bar{p}_K, \bar{q}_K) = \frac{1}{K} R_K - \frac{1}{2K} (p_0^\top A q_0 - p_K^\top A q_K). \tag{8}$$

If we are in the constrained case with bounded domains, then the last term in (8) is bounded, so the behavior of the duality gap is controlled by the growth of the total regret $R_K$.

**Bound on regret.**    If both players follow the alternating mirror descent algorithm with smooth convex regularizers, then we can bound the total regret $R_K$ as in Theorem 3.2 below. The proof uses the interpretation of alternating mirror descent as an alternating discretization of the skew-gradient flow, and a relation between the total regret and the change in the modified energy function, as we explain in Section 4.1.3. Smoothness of the regularizers allow us to control the discretization error, which translates to a bound on the regret. We provide the proof of Theorem 3.2 in Section C.1.2.

Here $\|\cdot\|$ is an arbitrary norm. We define the operator norm of $\nabla^3 \phi(x)$ (which is a 3-tensor) by $\|\nabla^3 \phi(x)\|_{\mathrm{op}} = \sup_{\|v\|=1} |\nabla^3 \phi(x)[v, v, v]|$. Let $\alpha_{\max}$ be the maximum singular value of $A$.

**Theorem 3.2.** *Assume $\mathcal{P}$ and $\mathcal{Q}$ are bounded, so $\|p\| \leq M$ and $\|q\| \leq M$ for all $p \in \mathcal{P}$, $q \in \mathcal{Q}$, for some $0 < M < \infty$. Assume the dual functions $\phi^*$ and $\psi^*$ are $M$-smooth of order $3$, which means $\|\nabla^3 \phi^*(\nabla \phi(p))\|_{\mathrm{op}} \leq M$ and $\|\nabla^3 \psi^*(\nabla \psi(q))\|_{\mathrm{op}} \leq M$ for all $p \in \mathcal{P}$, $q \in \mathcal{Q}$. Suppose both players follow alternating mirror descent (1) with step size $\eta > 0$ from $(p_0, q_0) \in \mathcal{P} \times \mathcal{Q}$. Then the total regret at iteration $K$ with respect to any strategy $(p, q) \in \mathcal{P} \times \mathcal{Q}$ is bounded by:*

$$R_K(p, q) \leq \frac{D_\phi(p, p_0) + D_\psi(q, q_0)}{\eta} + \frac{4 \alpha_{\max}^3 M^4}{3} \eta^2 K.$$

*In particular, suppose we start from $(p_0, q_0)$ with $\sup_{p \in \mathcal{P}} D_\phi(p, p_0) < \infty$ and $\sup_{q \in \mathcal{Q}} D_\psi(q, q_0) < \infty$.*
*Given a horizon $K$, with step size $\eta = \Theta(K^{-1/3})$, the total regret is*

$$R_K = O(K^{1/3}).$$

In the Euclidean case, Theorem 3.2 recovers [2, Theorem 2]. Theorem 3.2 also holds when $\mathcal{P}$, $\mathcal{Q}$ are probability simplices with entropy regularizers, if we start from $p_0, q_0$ with full support.

By Lemma 3.1 and Theorem 3.2, we have the following corollary (proof is in Section C.1.3).

**Corollary 3.3.** *Assume the same assumption as in Theorem 3.2. If both players follow the alternating mirror descent (1) with step size $\eta = \Theta(K^{-1/3})$, then the duality gap of the average iterates decays as $\mathsf{dg}(\bar{p}_K, \bar{q}_K) = O(K^{-2/3})$.*

Compare the result above with the simultaneous mirror descent algorithm for min-max games, which has the classical $O(K^{-1/2})$ convergence in the duality gap of the average iterates, thus highlighting the advantage of using the alternating mirror descent algorithm (at the cost of a third-order smoothness assumption for the analysis).

## 4 Skew-Gradient Flow and Discretization

In this section we discuss the skew-gradient flow and its discretization. We apply this to analyze the alternating mirror descent algorithm in the dual space $\mathbb{R}^m \times \mathbb{R}^n$.

Recall in the dual space, we have the dual functions of the convex regularizers $f := \phi^* \colon \mathbb{R}^m \to \mathbb{R}$ and $g := \psi^* \colon \mathbb{R}^n \to \mathbb{R}$. We define the **energy function** $H \colon \mathbb{R}^{m+n} \to \mathbb{R}$ by:

$$H(x, y) = f(x) + g(y)$$

for all $z = (x, y) \in \mathbb{R}^m \times \mathbb{R}^n$. Note $f = \phi^*$ and $g = \psi^*$ are convex, being convex conjugate functions. Furthermore, since we assume $\phi$ and $\psi$ are strictly convex, $f$ and $g$ are differentiable. Therefore, $H$ is a differentiable convex function.

Let $J \in \mathbb{R}^{(m+n) \times (m+n)}$ be the skew-symmetric matrix:

$$J = \begin{pmatrix} 0 & A \\ -A^\top & 0 \end{pmatrix}$$

where recall $A \in \mathbb{R}^{m \times n}$ is the payoff matrix for the min-max game.

We consider the **skew-gradient flow** generated by the energy function $H$ and the skew-symmetric matrix $J$, which is the solution to the differential equation:

$$\dot{Z}_t = -J \nabla H(Z_t) \tag{9}$$

starting from an arbitrary $Z_0 \in \mathbb{R}^{m+n}$. If we write $Z_t = (X_t, Y_t)$, then the components follow the skew-gradient flow dynamics as in (4):

$$\dot{X}_t = -A \nabla g(Y_t)$$
$$\dot{Y}_t = A^\top \nabla f(X_t).$$

A feature of the skew-gradient flow is that since the velocity is orthogonal to the gradient of the energy function, the skew-gradient flow preserves the energy function. (In contrast, recall the usual gradient flow $\dot{Z}_t = -\nabla H(Z_t)$ decreases the energy function.)

**Lemma 4.1.** *Along the skew-gradient flow (9), $H(Z_t) = H(Z_0)$ for all $t \geq 0$.*

*Proof.* We can check that the energy function $H(Z_t)$ has zero time derivative along the flow (9):

$$\frac{d}{dt} H(Z_t) = \langle \nabla H(Z_t), \dot{Z}_t \rangle = -\langle \nabla H(Z_t), J \nabla H(Z_t) \rangle = 0 \tag{10}$$

where the last equality above holds because $J = -J^\top$, so it defines a zero quadratic form. $\square$

Some of our results below hold for general skew-symmetric matrix $J$ (not necessarily in block structure). For simplicity, we focus on the separable case above for the min-max game application.

In continuous time, the skew-gradient flow dynamics (9) preserves the energy function. In discrete time, the behavior of the energy function can vary depending on the discretization method used.

**Forward Discretization.** A forward discretization of skew-gradient flow corresponds to the two players performing the simultaneous mirror descent algorithm for constrained min-max game. Since the energy function is convex, it is monotonically increasing along the forward method, and it is increasing exponentially fast if $H$ is strongly convex. See Section B.1 for details.

**Backward Discretization.** A backward discretization of skew-gradient flow corresponds to the two players performing the simultaneous proximal mirror descent algorithm for constrained min-max game. Since the energy function is convex, it is monotonically decreasing along the backward method, and it is decreasing exponentially fast if $H$ is strongly convex. See Section B.2 for details.

## 4.1 Alternating Discretization of Skew-Gradient Flow

The alternating discretization of the skew-gradient flow performs the updates one component at a time, using the forward method for the first component $(x)$ and the backward method for the second component $(y)$, resulting in the update equation:

$$x_{k+1} = x_k - \eta A \nabla_y H(x_{k+1}, y_k)$$
$$y_{k+1} = y_k + \eta A^\top \nabla_x H(x_{k+1}, y_k).$$

Since the energy function $H(x, y) = f(x) + g(y)$ is separable, this yields an explicit update equation:

$$x_{k+1} = x_k - \eta A \nabla g(y_k) \tag{11a}$$
$$y_{k+1} = y_k + \eta A^\top \nabla f(x_{k+1}). \tag{11b}$$

For the min-max game application, this corresponds to the alternating mirror descent update in the dual space (3).

In contrast to either the forward or the backward discretization, this alternating discretization tracks the continuous-time dynamics more closely, and thus we can bound the deviation in the energy function better. To explain our result, we introduce the notion of modified energy function.

### 4.1.1 Modified Energy Function

Given a step size $\eta > 0$, we define the **modified energy** function $H_\eta \colon \mathbb{R}^{m+n} \to \mathbb{R}$ by:

$$H_\eta(z) = H(z) - \frac{\eta}{2} \langle \nabla f(x), A \nabla g(y) \rangle \tag{12a}$$
$$= f(x) + g(y) - \frac{\eta}{2} \langle \nabla f(x), A \nabla g(y) \rangle \tag{12b}$$

for $z = (x, y) \in \mathbb{R}^{m+n}$. Note that when $f$ and $g$ are quadratic functions, this recovers the definition of the modified energy function in [2].

Recall the Bregman divergence $D_H(z', z) = H(z') - H(z) - \langle \nabla H(z), z' - z \rangle$ is not necessarily symmetric. We define the **Bregman commutator** $C_H \colon \mathbb{R}^{m+n} \times \mathbb{R}^{m+n} \to \mathbb{R}$ as a measure of the non-commutativity of Bregman divergence:

$$C_H(z', z) = \tfrac{1}{2}(D_H(z', z) - D_H(z, z'))$$
$$= H(z') - H(z) - \tfrac{1}{2} \langle \nabla H(z') + \nabla H(z), z' - z \rangle.$$

Observe that when $H(z) = \frac{1}{2} z^\top M z$ is a quadratic function (for any positive definite matrix $M$), then $D_H(z', z) = \frac{1}{2}(z' - z)^\top M(z' - z) = D_H(z, z')$ is symmetric, and thus $C_H(z', z) = 0$. Conversely, if the Bregman commutator vanishes: $C_H(z', z) = 0$, then $H$ must be a quadratic function; see Lemma A.1 in Section A.2.1. We can also bound the Bregman commutator under third-order smoothness; see Lemma A.2 in Section A.2.1.

We show that along the alternating method, the modified energy changes by precisely the Bregman commutator; we provide the proof in Section B.3.2.

**Lemma 4.2.** *Let $z_k = (x_k, y_k)$ evolve following the alternating method (11). Then for any $k \geq 0$,*

$$H_\eta(z_{k+1}) = H_\eta(z_k) + C_H(z_{k+1}, z_k). \tag{13}$$

As a corollary, if $H$ is quadratic—in which case the Bregman commutator vanishes—then the modified energy function is conserved along the alternating method. This recovers the main result of [2] for the unconstrained min-max game. We provide the proof of Corollary 4.3 in Section B.3.1.

**Corollary 4.3.** *Suppose $H$ is a quadratic function. Along the alternating method* (3)*, for any $k \geq 0$:*

$$H_\eta(z_k) = H_\eta(z_0).$$

#### 4.1.2 Bound under Third-Order Smoothness

Under smoothness assumptions on $H$, we can deduce the following bound on the modified energy function. Let $\|\cdot\|$ be an arbitrary norm in $\mathbb{R}^{m+n}$ with dual norm $\|\cdot\|_*$. Recall we say $H$ is $L$-Lipschitz if $|H(z') - H(z)| \leq L\|z' - z\|$ for all $z, z' \in \mathbb{R}^{m+n}$; equivalently, $\|\nabla H(z)\|_* \leq L$. We say $H$ is $M$-smooth of order 3 if $H$ is three-times differentiable, and $\|\nabla^3 H(z)\|_{\mathrm{op}} \leq M$ for all $z \in \mathbb{R}^{m+n}$, where $\|\cdot\|_{\mathrm{op}}$ is the operator norm. Then we have the following bound. We provide the proof of Theorem 4.4 in Section B.3.3. We note the following result holds without assuming convexity of $H$.

**Theorem 4.4.** *Assume that $H$ is $L_1$-Lipschitz and $L_3$-smooth of order* 3*. Let $\alpha_{\max}$ be the maximum singular value of $A$. Along the alternating method* (11)*, for any $\eta \geq 0$ and at any iteration $k \geq 0$:*

$$|H_\eta(z_k) - H_\eta(z_0)| \leq \tfrac{1}{12} \alpha_{\max}^3 \, L_3 \, L_1^3 \, \eta^3 \, k.$$

As an example, the function $H(z) = \log \sum_{i=1}^m e^{z_i} + \log \sum_{j=1}^n e^{z_{m+j}}$ satisfies the assumptions of Theorem 4.4 with respect to $\|\cdot\|_\infty$-norm; see Example A.3 in Section A.2.1. In our min-max game application, this function arises from using the entropy regularizer on the probability simplex.

#### 4.1.3 Regret of Alternating Mirror Descent in terms of Modified Energy

We now consider the constrained min-max game application where $H(x, y) = f(x) + g(y)$ with $f = \phi^*$ and $g = \psi^*$. Given $(p, q) \in \mathcal{P} \times \mathcal{Q}$, let $z = (x, y) \in \mathbb{R}^{m+n}$ be the dual variables, where $x = \nabla\phi(p)$ and $y = \nabla\psi(q)$. Then we can write the cumulative regret of alternating mirror descent in terms of the difference of the modified energy. We provide the proof of Theorem 4.5 in Section B.3.4.

**Theorem 4.5.** *Let $(p_k, q_k)$ evolve following the alternating mirror descent algorithm* (1) *with any step size $\eta > 0$, and let $z_k = (x_k, y_k)$ be the dual variables. We can write the cumulative regret of the two players with respect to any $(p, q) \in \mathcal{P} \times \mathcal{Q}$ as:*

$$R_K(p, q) = \tfrac{1}{\eta} \left( D_H(z_0, z) - D_H(z_K, z) + H_\eta(z_K) - H_\eta(z_0) \right).$$

Since $H$ is convex, we can further bound $D_H(z_K, z) \geq 0$, so from the above we get:

$$R_K(p, q) \leq \tfrac{1}{\eta} \left( D_H(z_0, z) + H_\eta(z_K) - H_\eta(z_0) \right).$$

The first term in the bound above is a constant that only depends on the initial point. If we assume the domains are bounded, then this first term is also bounded. Thus, to control the cumulative regret, we need to bound the increase in the modified energy, which we can do via Theorem 4.4 under third-order smoothness. Completing this step yields the proof of Theorem 3.2; see Section C.1.2.

## 5 Discussion

In this paper we study the alternating mirror descent algorithm for constrained min-max games, and showed that it achieves a better regret bound than the classical simultaneous mirror descent. Our results extend the findings of [2] from the unconstrained to the constrained setting. We have utilized an interpretation of alternating mirror descent as an alternating discretization of the skew-gradient flow in the dual space, and linked the total regret of the players with the growth of the modified energy function. Our analysis highlights the connections between min-max games and Hamiltonian structures, which helps pave the way toward a closer interplay between classical numerical methods and modern algorithmic questions motivated by machine learning applications.

Our results leave many interesting open questions. First, our bound in Theorem 4.4 grows with the number of iterations, and does not yield constant regret as in the unconstrained case. In the special case of identity payoff matrix, in which the skew-gradient flow dynamics has a Hamiltonian and symplectic structure, we conjecture that for sufficiently nice energy functions, the iterates of the

alternating method stay uniformly bounded, as in the unconstrained setting. In the Appendix, we provide some empirical evidence supporting this conjecture. Resolving this question requires making concrete classical bounds from numerical methods, which may be of independent interest.

Second, we have focused on the simple case of two-player bilinear game, which already presents non-trivial behavior. It would be interesting to understand the more general setting of multi-player games with general payoffs, in which the notion of "alternating" play can take many possible forms. It would also be interesting to study the asynchronous setting in which the players make their moves in decentralized or randomized fashions. This will help us understand and control the global behavior of multi-agent systems and how they emerge from simple local or individual strategies.

## Acknowledgements

We thank the reviewers for helpful comments and discussions on the paper. This research/project is supported in part by the National Research Foundation, Singapore and DSO National Laboratories under its AI Singapore Program (AISG Award No: AISG2-RP-2020-016), NRF 2018 Fellowship NRF-NRFF2018-07, NRF2019-NRF-ANR095 ALIAS grant, grant PIESGP-AI-2020-01, AME Programmatic Fund (Grant No.A20H6b0151) from the Agency for Science, Technology and Research (A*STAR) and Provost's Chair Professorship grant RGEPPV2101.

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
