# A Helper Lemmas

## A.1 Properties of Bregman Divergence

Let $f \colon \mathbb{R}^m \to \mathbb{R}$ be a differentiable function. Recall the *Bregman divergence* of $f$ is given by:

$$D_f(x', x) = f(x') - f(x) - \langle \nabla f(x), x' - x \rangle$$

for all $x, x' \in \mathbb{R}^m$. The Bregman divergence is in general not symmetric: $D_f(x', x) \neq D_f(x, x')$.

We recall the three-point identity for Bregman divergence:

$$D_f(x, y) - D_f(x, z) - D_f(z, y) = \langle \nabla f(z) - \nabla f(y), x - z \rangle.$$

Recall we say $f$ is $\alpha$-*strongly convex* with respect to a norm $\| \cdot \|$ if

$$D_f(x', x) \geq \frac{\alpha}{2} \|x' - x\|^2.$$

For example, if $f(x) = \frac{1}{2} x^\top M x$ is a quadratic function defined by a symmetric positive definite matrix $M \in \mathbb{R}^{m \times m}$, then $f$ is strongly convex in the $\ell_2$-norm with strong convexity constant equal to the smallest eigenvalue of $M$. If $f(x) = \sum_{i=1}^m x_i \log x_i$ is the negative entropy function defined on the simplex $\Delta_m \subset \mathbb{R}^m$, then $f$ is 1-strongly convex in the $\ell_1$-norm.

Recall we say $f$ is $\alpha$-*gradient dominated* with respect to the dual norm $\| \cdot \|_*$ if

$$\|\nabla f(x)\|_*^2 \geq 2\alpha(f(x) - \min f)$$

for all $x \in \mathbb{R}^m$, where $\min f = \min_x f(x)$ is the minimum value of $f$.

We recall that strong convexity implies gradient domination with the same constant.

## A.2 Properties of Bregman commutator

The *Bregman commutator* of $f$ is the function $C_f \colon \mathbb{R}^m \times \mathbb{R}^m \to \mathbb{R}$ that measures the failure of the Bregman divergence to be symmetric:

$$C_f(x', x) = \frac{1}{2} \big( D_f(x', x) - D_f(x, x') \big)$$

$$= f(x') - f(x) - \frac{1}{2} \langle \nabla f(x') + \nabla f(x), x' - x \rangle.$$

By definition, the Bregman commutator is antisymmetric:

$$C_f(x, x') = -C_f(x', x).$$

For example, if $f$ is a quadratic function, then the Bregman divergence is also a quadratic function, which is symmetric, and hence the Bregman commutator vanishes: $C_f(x, x') = 0$ for all $x, x' \in \mathbb{R}^m$. The converse also holds: If the Bregman commutator vanishes, then the function must be a quadratic.

**Lemma A.1.** *If $C_f(x', x) = 0$ for all $x', x \in \mathbb{R}^m$, then $f$ is a quadratic function.*

*Proof.* By assumption, we have $D_f(x', x) = D_f(x, x')$ for all $x, x' \in \mathbb{R}^m$. It suffices to argue that $\tilde{f}(x) := f(x) - f(0) - \langle x, \nabla f(0) \rangle$ is a quadratic function, for then $f$ is also quadratic. Since this is a linear transformation, it does not affect the Bregman divergence, and thus by assumption we have $D_{\tilde{f}}(x', x) = D_{\tilde{f}}(x, x')$ for all $x, x' \in \mathbb{R}^m$. Note that by definition, $\tilde{f}(0) = 0$ and $\nabla \tilde{f}(0) = 0$. Now, the relation $D_{\tilde{f}}(x, 0) = D_{\tilde{f}}(0, x)$ implies that $\tilde{f}(x) = \frac{1}{2} \langle \nabla \tilde{f}(x), x \rangle$. Thus, $D_{\tilde{f}}(x', x) = D_{\tilde{f}}(x, x')$ implies that $\langle \nabla \tilde{f}(x), x' \rangle = \langle \nabla \tilde{f}(x'), x \rangle$ for all $x, x' \in \mathbb{R}^m$. This in turn implies, for any $x, x'$, and $y$,

$$\langle \nabla \tilde{f}(x + x'), y \rangle = \langle \nabla \tilde{f}(y), x + x' \rangle$$
$$= \langle \nabla \tilde{f}(y), x \rangle + \langle \nabla \tilde{f}(y), x' \rangle$$
$$= \langle \nabla \tilde{f}(x), y \rangle + \langle \nabla \tilde{f}(x'), y \rangle$$
$$= \langle \nabla \tilde{f}(x) + \nabla \tilde{f}(x'), y \rangle.$$

This shows that $\nabla \tilde{f}$ is linear, which means $\tilde{f}$ is quadratic, and hence $f$ is also quadratic. $\square$

### A.2.1  Bound on Bregman commutator

Recall $f$ is $M$-smooth of order 3 if $f\colon \mathbb{R}^m \to \mathbb{R}$ is three-times differentiable and for all $x \in \mathbb{R}^m$:

$$\|\nabla^3 f(x)\|_{\mathrm{op}} \leq M$$

where $\|\cdot\|_{\mathrm{op}}$ is the operator norm; concretely, this means for all $x \in \mathbb{R}^m$ and $v \in \mathbb{R}^m$ with $\|v\| = 1$,

$$|\nabla^3 f(x)[v, v, v]| \leq M.$$

Equivalently, $\nabla^2 f$ is $M$-Lipschitz.

We have the following bound on Bregman commutator under third-order smoothness.

**Lemma A.2.** *Assume $f$ is $M$-smooth of order 3 with respect to a norm $\|\cdot\|$. Then for all $x', x \in \mathbb{R}^m$:*

$$|C_f(x, x')| \leq \frac{M}{12}\|x - x'\|^3. \tag{14}$$

*Proof.* Recall the Taylor expansion

$$r(1) = r(0) + \dot{r}(0) + \int_0^1 (1 - t)\ddot{r}(t)\,dt$$

for any twice-differentiable function $r\colon [0, 1] \to \mathbb{R}$. Applying this with $r(t) = f(x + t(x' - x))$ gives

$$f(x') = f(x) + \langle \nabla f(x), x' - x \rangle + \int_0^1 (1 - t)\langle (x' - x), \nabla^2 f(x + t(x' - x))\,(x' - x)\rangle\,dt,$$

or equivalently,

$$D_f(x', x) = \left\langle (x' - x), \left( \int_0^1 (1 - t)\nabla^2 f(x + t(x' - x))\,dt \right)(x' - x) \right\rangle.$$

Similarly,

$$D_f(x, x') = \left\langle (x' - x), \left( \int_0^1 t\nabla^2 f(x + (1 - t)(x' - x))\,dt \right)(x' - x) \right\rangle.$$

Combining the two equations above, we obtain

$$C_f(x', x) = \frac{1}{2}\big(D_f(x', x) - D_f(x, x')\big)$$

$$= \frac{1}{2}\left\langle (x' - x), \left( \int_0^1 (1 - 2t)\nabla^2 f(x + t(x' - x))\,dt \right)(x' - x) \right\rangle. \tag{15}$$

For $0 \leq t \leq 1$, let $s = t - \frac{1}{2}$ and $s' = -s$. We denote $x_s \equiv x + (s + \frac{1}{2})(x' - x)$. Then we can write the integral above as

$$\frac{1}{2}\int_0^1 (1 - 2t)\nabla^2 f(x_{t - \frac{1}{2}})\,dt = \int_{-\frac{1}{2}}^{\frac{1}{2}} (-s)\nabla^2 f(x_s)\,ds$$

$$= \int_{-\frac{1}{2}}^0 (-s)\nabla^2 f(x_s)\,ds + \int_0^{\frac{1}{2}} (-s)\nabla^2 f(x_s)\,ds$$

$$= \int_0^{\frac{1}{2}} s'\nabla^2 f(x_{-s'})\,ds' + \int_0^{\frac{1}{2}} (-s)\nabla^2 f(x_s)\,ds$$

$$= \int_0^{\frac{1}{2}} s\left(\nabla^2 f(x_{-s}) - \nabla^2 f(x_s)\right)ds. \tag{16}$$

By the mean-value theorem and using $x_{-s} - x_s = -2s(x' - x)$, we can write

$$\nabla^2 f(x_{-s}) - \nabla^2 f(x_s) = \int_0^1 \nabla^3 f(x_s + u(x_{-s} - x_s))[x_{-s} - x_s]\,du$$

$$= -2s \int_0^1 \nabla^3 f(x_s + u(x_{-s} - x_s))[x' - x]\,du.$$

Plugging this in to (16), we obtain

$$\frac{1}{2} \int_0^1 (1 - 2t)\nabla^2 f(x_{t-\frac{1}{2}})\, dt = -2 \int_0^{\frac{1}{2}} \int_0^1 s^2 \nabla^3 f(x_s + u(x_{-s} - x_s))[x' - x]\, du\, ds.$$

Plugging this in to (15), we then obtain

$$C_f(x', x) = -2 \int_0^{\frac{1}{2}} \int_0^1 s^2 \nabla^3 f(x_s + u(x_{-s} - x_s))[x' - x, x' - x, x' - x]\, du\, ds.$$

Using the third-order smoothness property of $f$, we can then bound

$$|C_f(x', x)| \le 2 \int_0^{\frac{1}{2}} \int_0^1 s^2 \big| \nabla^3 f(x_s + u(x_{-s} - x_s))[x' - x, x' - x, x' - x] \big|\, du\, ds$$

$$\le 2 \int_0^{\frac{1}{2}} \int_0^1 s^2 M \|x' - x\|^3\, du\, ds$$

$$= 2M\|x' - x\|^3 \cdot \frac{1}{3}\left(\frac{1}{2}\right)^3$$

$$= \frac{M\|x' - x\|^3}{12}$$

as desired. $\qquad\square$

**Example A.3.** *Consider $f(x) = \log \sum_{i=1}^m e^{x_i}$. This is the log-partition function of an exponential family distribution $p_x$ over a finite set $[m] := \{1, \dots, m\}$ with $p_x(i) = e^{x_i - f(x)}$. Then $\nabla^3 f(x)$ is the third-order covariance, i.e., for all $v \in \mathbb{R}^m$,*

$$\nabla^3 f(x)[v, v, v] = \mathrm{Cov}_{I \sim p_x}^3(v_I) = \mathbb{E}_{I \sim p_x}[(v_I - \bar{v})^3]$$

*where $\bar{v} = \mathbb{E}_{I \sim p_x}[v_I] = \sum_{i=1}^m v_i p_x(i)$. Suppose $\|v\|_\infty = \max_i |v_i| = 1$, so $|\bar{v}| \le 1$, and $|v_i - \bar{v}| \le 2$. Then*

$$|\nabla^3 f(x)[v, v, v]| \le \mathbb{E}_{I \sim p_x}[|v_I - \bar{v}|^3] \le \mathbb{E}_{I \sim p_x}[2^3] = 8.$$

*This shows that $f$ is 8-smooth of order 3 with respect to the $\ell_\infty$-norm $\|\cdot\|_\infty$.*

# B Discretization of Skew-Gradient Flow

## B.1 Forward Discretization of Skew-Gradient Flow

Consider the forward method to discretize the skew-gradient flow (9) with step size $\eta > 0$:

$$z_{k+1} = z_k - \eta J \nabla H(z_k). \tag{17}$$

In terms of the components $z_k = (x_k, y_k)$, this corresponds to the simultaneous forward method:

$$x_{k+1} = x_k - \eta A \nabla g(y_k)$$

$$y_{k+1} = y_k + \eta A^\top \nabla f(x_k).$$

For the min-max game application when $f = \nabla \phi^*$ and $g = \psi^*$, this corresponds to the two players following the simultaneous mirror descent algorithm:

$$p_{k+1} = \arg\min_{p \in \mathcal{P}} \left\{ p^\top A q_k + \frac{1}{\eta} D_\phi(p, p_k) \right\}$$

$$q_{k+1} = \arg\min_{q \in \mathcal{Q}} \left\{ -p_k^\top A q + \frac{1}{\eta} D_\psi(q, q_k) \right\}.$$

We show the following properties of the forward method. Here let $\|\cdot\| = \|\cdot\|_2$ be the $\ell_2$-norm. Recall $H$ is $L$-smooth if $\nabla H(z)$ is $L$-Lipschitz:

$$\|\nabla H(z) - \nabla H(z')\|_* \le L\|z - z'\|.$$

Equivalently, $\|\nabla^2 H(z)\|_{\mathrm{op}} \le L$ where $\|\cdot\|_{\mathrm{op}}$ is the operator norm, and $\nabla^2 H(z)$ is the Hessian matrix of second derivatives.

**Theorem B.1.** *Let $z_k$ evolve following the forward method* (17). *We have the following properties:*

1. *If $H$ is convex, then $H(z_{k+1}) \geq H(z_k)$.*

2. *Assume $m = n$, and assume $A \in \mathbb{R}^{n \times n}$ is invertible with smallest singular value $\alpha_{\min} > 0$. Assume $H$ is $\mu$-strongly convex, and let $z^* = \arg\min_z H(z)$. Then:*

$$H(z_k) - H(z^*) \geq (1 + \eta^2 \alpha_{\min}^2 \mu^2)^k \left( H(z_0) - H(z^*) \right).$$

3. *Assume $H$ is convex and $L_2$-smooth. Let $\alpha_{\max}$ be the maximum singular value of $A$. Then:*

$$H(z_k) - H(z^*) \leq (1 + \eta^2 \alpha_{\max}^2 L^2)^k \left( H(z_0) - H(z^*) \right)$$

The properties above say that if the energy function is convex, it is monotonically increasing along the forward method; furthermore, it is increasing exponentially fast if $H$ is strongly convex. On the other hand, if $H$ is convex and smooth, then it is increasing at most exponentially fast.

We note that while the assumptions in Theorem B.1 are stated with respect to the $\ell_2$-norm $\| \cdot \|_2$, since all norms in $\mathbb{R}^{m+n}$ are equivalent, the results also hold for any other norm $\| \cdot \|$ (with smoothness and strong convexity constants that may have dimension dependence). Here for simplicity we provide the proof for the $\ell_2$-norm. An explicit calculation on a quadratic example shows that the rates in Theorem B.1 are tight; see Example B.2.

*Proof of Theorem B.1.*      1. First assume $H$ is convex. By Jensen's inequality,

$$H(z_{k+1}) - H(z_k) \geq \langle \nabla H(z_k), z_{k+1} - z_k \rangle = \eta \langle \nabla H(z_k), J \nabla H(z_k) \rangle = 0.$$

This shows the forward method always increases the energy function: $H(z_{k+1}) \geq H(z_k)$.

2. Assume $m = n$, and $A \in \mathbb{R}^{n \times n}$ has smallest singular value $\alpha_{\min} > 0$. This implies $A^\top A \succeq \alpha_{\min}^2 I_n$ and $A A^\top \succeq \alpha_{\min}^2 I_n$, so $J^\top J \succeq \alpha_{\min}^2 I_{2n}$.

Assume further that $H$ is $\mu$-strongly convex. This implies $H$ is $\mu$-gradient dominated:

$$\|\nabla H(z)\|_2^2 \geq 2\mu(H(z) - H(z^*))$$

for all $z \in \mathbb{R}^{2n}$. Then by the $\mu$-strong convexity of $H$, along the forward method,

$$
\begin{aligned}
H(z_{k+1}) - H(z_k) &\geq \langle \nabla H(z_k), z_{k+1} - z_k \rangle + \frac{\mu}{2} \|z_{k+1} - z_k\|_2^2 \\
&= -\eta \langle \nabla H(z_k), J \nabla H(z_k) \rangle + \frac{\eta^2 \mu}{2} \|J \nabla H(z_k)\|_2^2 \\
&= 0 + \frac{\eta^2 \mu}{2} \langle \nabla H(z_k), (J^\top J) \nabla H(z_k) \rangle \\
&\geq \frac{\eta^2 \alpha_{\min}^2 \mu}{2} \|\nabla H(z_k)\|_2^2 \\
&\geq \eta^2 \alpha^2 \mu^2 (H(z_k) - H(z^*))
\end{aligned}
$$

This implies $H(z_{k+1}) - H(z^*) \geq (1 + \eta^2 \alpha_{\min}^2 \mu^2)(H(z_k) - H(z^*))$. Therefore, the forward method increases the energy function exponentially fast:

$$H(z_k) - H(z^*) \geq (1 + \eta^2 \alpha_{\min}^2 \mu^2)^k (H(z_0) - H(z^*)).$$

3. Since $A$ has maximum singular value $\alpha_{\max}$, we have $A^\top A \preceq \alpha_{\max}^2 I_n$ and $A A^\top \preceq \alpha_{\max}^2 I_m$, so $J^\top J \preceq \alpha_{\max}^2 I_{m+n}$.

Since $H$ is convex and $L_2$-smooth, we have for all $z \in \mathbb{R}^{m+n}$ (see e.g. [22]):

$$H(z) \geq H(z^*) + \frac{1}{2L_2} \|\nabla H(z)\|_2^2.$$

Then by the $L_2$-smoothness of $H$, along the forward method,

$$H(z_{k+1}) - H(z_k) \le \langle \nabla H(z_k), z_{k+1} - z_k \rangle + \frac{L_2}{2}\|z_{k+1} - z_k\|_2^2$$

$$= -\eta \langle \nabla H(z_k), J\nabla H(z_k) \rangle + \frac{\eta^2 L_2}{2}\|J\nabla H(z_k)\|_2^2$$

$$= 0 + \frac{\eta^2 L_2}{2} \langle \nabla H(z_k), (J^\top J) \nabla H(z_k) \rangle$$

$$\le \frac{\eta^2 \alpha_{\max}^2 L_2}{2}\|\nabla H(z_k)\|_2^2$$

$$\le \eta^2 \alpha_{\max}^2 L_2^2 (H(z_k) - H(z^*))$$

This implies $H(z_{k+1}) - H(z^*) \le (1 + \eta^2 \alpha_{\max}^2 L_2^2)(H(z_k) - H(z^*))$. Therefore, along the forward method, the energy function increases at most exponentially fast:

$$H(z_k) - H(z^*) \le (1 + \eta^2 \alpha_{\max}^2 L_2^2)^k (H(z_0) - H(z^*)).$$

$\square$

In Section C.3 we analyze the total regret of the forward method for the min-max game application, and show the duality gap of the average iterate decays at an $O(K^{-1/2})$ rate; see Theorem C.2.

### B.1.1 Quadratic example

We can check that in the quadratic case, the rates in Theorem B.1 are tight in terms of dependence on step size $\eta$.

**Example B.2** (Quadratic). *Suppose $f(x) = \frac{\beta}{2}\|x\|_2^2$ and $g(y) = \frac{\gamma}{2}\|y\|_2^2$ for some $\beta, \gamma > 0$. Then for $z = (x, y)$, the energy function is $H(z) = \frac{1}{2}z^\top M z$ where $M = \begin{pmatrix} \beta I_m & 0 \\ 0 & \gamma I_n \end{pmatrix}$, where $I_m \in \mathbb{R}^{m \times m}$ and $I_n \in \mathbb{R}^{n \times n}$ are the identity matrices. Here $H$ is $\mu$-strongly convex and $L$-smooth, where $\mu = \min\{\beta, \gamma\}$ and $L = \max\{\beta, \gamma\}$.*

*The forward method* (17) *becomes*

$$z_{k+1} = z_k + \eta J M z_k = (I_{m+n} - \eta J M)z_k.$$

*Then*

$$H(z_{k+1}) = \frac{1}{2}z_{k+1}^\top M z_{k+1}$$

$$= \frac{1}{2}z_k^\top (I_{m+n} - \eta M^\top J^\top)M(I_{m+n} - \eta J M)z_k$$

$$= \frac{1}{2}z_k^\top \begin{pmatrix} I_m & -\eta\beta A \\ \eta\gamma A^\top & I_n \end{pmatrix} \begin{pmatrix} \beta I_m & 0 \\ 0 & \gamma I_n \end{pmatrix} \begin{pmatrix} I_m & \eta\gamma A \\ -\eta\beta A^\top & I_n \end{pmatrix} z_k$$

$$= \frac{1}{2}z_k^\top \begin{pmatrix} \beta I_m + \eta^2\beta^2\gamma AA^\top & 0 \\ 0 & \gamma I_n + \eta^2\beta\gamma^2 A^\top A \end{pmatrix} z_k.$$

*If $m \ne n$, then one of $AA^\top$, $A^\top A$ will be low-rank and thus have a zero eigenvalue, so*

$$H(z_{k+1}) \ge \frac{1}{2}z_k^\top \begin{pmatrix} \beta I_m & 0 \\ 0 & \gamma I_n \end{pmatrix} z_k = H(z_k).$$

*Suppose $m = n$ and $A$ has smallest singular value $\alpha_{\min} > 0$, so $AA^\top \succeq \alpha^2 I_n$ and $A^\top A \succeq \alpha^2 I_n$. Then*

$$H(z_{k+1}) \ge \frac{1}{2}z_k^\top \begin{pmatrix} \beta I_m + \eta^2\beta^2\gamma\alpha_{\min}^2 I_m & 0 \\ 0 & \gamma I_n + \eta^2\beta\gamma^2\alpha_{\min}^2 I_n \end{pmatrix} z_k$$

$$= \frac{(1 + \eta^2\alpha_{\min}^2\beta\gamma)}{2}z_k^\top \begin{pmatrix} \beta I_m & 0 \\ 0 & \gamma I_n \end{pmatrix} z_k$$

$$= (1 + \eta^2\alpha_{\min}^2\beta\gamma)H(z_k).$$

*Note in this case the minimizer is $z^* = 0$ with $H(z^*) = 0$. Therefore,*

$$H(z_k) - H(z^*) \geq (1 + \eta^2 \alpha_{\min}^2 \beta \gamma)^k (H(z_0) - H(z^*)).$$

*Finally, let $\alpha_{\max}$ be the maximum singular value of $A$. Then*

$$
\begin{aligned}
H(z_{k+1}) &= \frac{1}{2} z_k^\top \begin{pmatrix} \beta I_m + \eta^2 \beta^2 \gamma A A^\top & 0 \\ 0 & \gamma I_n + \eta^2 \beta \gamma^2 A^\top A \end{pmatrix} z_k \\
&\leq \frac{1}{2} z_k^\top \begin{pmatrix} \beta I_m + \eta^2 \beta^2 \gamma \alpha_{\max}^2 I_m & 0 \\ 0 & \gamma I_n + \eta^2 \beta \gamma^2 \alpha_{\max}^2 I_n \end{pmatrix} z_k \\
&= \frac{(1 + \eta^2 \alpha_{\max}^2 \beta \gamma)}{2} z_k^\top \begin{pmatrix} \beta I_m & 0 \\ 0 & \gamma I_n \end{pmatrix} z_k \\
&= (1 + \eta^2 \alpha_{\max}^2 \beta \gamma) H(z_k).
\end{aligned}
$$

*Therefore,*

$$H(z_k) - H(z^*) \leq (1 + \eta^2 \alpha_{\max}^2 \beta \gamma)^k (H(z_0) - H(z^*)).$$

## B.2 Backward Discretization of Skew-Gradient Flow

Consider the forward method to discretize the skew-gradient flow (9) with step size $\eta > 0$:

$$z_{k+1} = z_k - \eta J \nabla H(z_{k+1}). \tag{18}$$

This is an implicit update, and may require some computation to solve in each iteration. If $H$ is smooth, then the update is well-defined for small enough step size.

In terms of the components $z_k = (x_k, y_k)$, this corresponds to the simultaneous backward method:

$$
\begin{aligned}
x_{k+1} &= x_k - \eta A \nabla g(y_{k+1}) \\
y_{k+1} &= y_k + \eta A^\top \nabla f(x_{k+1}).
\end{aligned}
$$

For the min-max game application when $f = \nabla \phi^*$ and $g = \psi^*$, this corresponds to the two players following the simultaneous proximal mirror descent algorithm:

$$
\begin{aligned}
p_{k+1} &= \arg\min_{p \in \mathcal{P}} \left\{ p^\top A q_{k+1} + \frac{1}{\eta} D_\phi(p, p_k) \right\} \\
q_{k+1} &= \arg\min_{q \in \mathcal{Q}} \left\{ -p_{k+1}^\top A q + \frac{1}{\eta} D_\psi(q, q_k) \right\}.
\end{aligned}
$$

We show the following properties of the backward method, which are the opposite of the properties of the forward method. Here let $\|\cdot\| = \|\cdot\|_2$ be the $\ell_2$-norm.

**Theorem B.3.** *Let $z_k$ evolve following the backward method* (18)*. We have the following properties:*

1. *If $H$ is $L_2$-smooth and $\eta < \frac{1}{\alpha_{\max} L_2}$, then the backward method is well-defined, i.e. the solution $z_{k+1}$ exists and is unique for any $z_k$.*

2. *If $H$ is convex, then $H(z_{k+1}) \leq H(z_k)$.*

3. *Assume $m = n$, and assume $A \in \mathbb{R}^{n \times n}$ is invertible with smallest singular value $\alpha_{\min} > 0$. Assume $H$ is $\mu$-strongly convex with respect, and let $z^* = \arg\min_z H(z)$. Then:*

$$H(z_k) - H(z^*) \leq \frac{H(z_0) - H(z^*)}{(1 + \eta^2 \alpha_{\min}^2 \mu^2)^k}.$$

4. *Assume $H$ is convex and $L_2$-smooth. Then:*

$$H(z_k) - H(z^*) \geq \frac{H(z_0) - H(z^*)}{(1 + \eta^2 \alpha_{\max}^2 L^2)^k}$$

As in Theorem B.1, the results in Theorem B.3 are stated in terms of the $\ell_2$-norm for simplicity, but also hold for any norm $\|\cdot\|$ in $\mathbb{R}^{m+n}$, at the cost of a potentially dimension-dependence constants. We note that we can also deduce Theorem B.3 by reversing the argument of Theorem B.1, since the backward method is the adjoint of the forward method (i.e. the inverse with negated step size $-\eta$). For completeness, we provide the full proof below. We can also check that in the quadratic case, the rates in Theorem B.3 are tight.

*Proof of Theorem B.3.*      1. We can write the update (18) as $(I + \eta J\nabla H)(z_{k+1}) = z_k$, where $I$ is the identity operator. The Jacobian matrix of the operator $I + \eta J\nabla H$ is $I_{m+n} + \eta J\nabla^2 H$, which has smallest singular value at least $1 - \eta\alpha_{\max}\|\nabla^2 H\|_{\mathrm{op}} \geq 1 - \eta\alpha_{\max}L_2$. We see that if $\eta < \frac{1}{\alpha_{\max}L_2}$, then the Jacobian matrix is non-singular, thus the operator $I + \eta J\nabla H$ is invertible, and hence $z_{k+1} = (I + \eta J\nabla H)^{-1}(z_k)$ exists and is uniquely defined.

2. Assume $H$ is convex. By Jensen's inequality,
$$H(z_{k+1}) - H(z_k) \leq \langle \nabla H(z_{k+1}), z_{k+1} - z_k \rangle = -\eta\langle \nabla H(z_{k+1}), J\nabla H(z_{k+1})\rangle = 0.$$
This shows the backward method always decreases the energy function: $H(z_{k+1}) \leq H(z_k)$.

3. Assume $m = n$, and $A \in \mathbb{R}^{n\times n}$ has a positive smallest singular value $\alpha_{\min} > 0$. This implies $A^\top A \succeq \alpha_{\min}^2 I_n$ and $AA^\top \succeq \alpha_{\min}^2 I_n$, so $J^\top J \succeq \alpha_{\min}^2 I_{2n}$.

Assume further that $H$ is $\mu$-strongly convex, which implies $H$ is $\mu$-gradient dominated:
$$\|\nabla H(z)\|_2^2 \geq 2\mu(H(z) - H(z^*))$$
for all $z \in \mathbb{R}^{2n}$. Then by the $\mu$-strong convexity of $H$, along the backward method,
$$H(z_{k+1}) - H(z_k) \leq \langle \nabla H(z_{k+1}), z_{k+1} - z_k \rangle - \frac{\mu}{2}\|z_{k+1} - z_k\|_2^2$$
$$= -\eta\langle \nabla H(z_{k+1}), J\nabla H(z_{k+1})\rangle - \frac{\eta^2\mu}{2}\|J\nabla H(z_{k+1})\|_2^2$$
$$\leq -\frac{\eta^2\alpha_{\min}^2\mu}{2}\|\nabla H(z_{k+1})\|_2^2$$
$$\leq -\eta^2\alpha_{\min}^2\mu^2(H(z_{k+1}) - H(z^*))$$
This implies
$$H(z_{k+1}) - H(z^*) \leq \frac{H(z_k) - H(z^*)}{1 + \eta^2\alpha_{\min}^2\mu^2}.$$
Therefore, the backward method decreases the energy function exponentially fast:
$$H(z_k) - H(z^*) \leq \frac{H(z_0) - H(z^*)}{(1 + \eta^2\alpha_{\min}^2\mu^2)^k}.$$

4. Since $A$ has maximum singular value $\alpha_{\max} > 0$, we have $A^\top A \preceq \alpha_{\max}^2 I_n$ and $AA^\top \preceq \alpha_{\max}^2 I_m$, so $J^\top J \preceq \alpha_{\max}^2 I_{m+n}$.

Since $H$ is convex and $L$-smooth, we have for all $z \in \mathbb{R}^{m+n}$:
$$H(z) \geq H(z^*) + \frac{1}{2L}\|\nabla H(z)\|_2^2.$$

Then by the $L_2$-smoothness of $H$, along the backward method,
$$H(z_{k+1}) - H(z_k) \geq \langle \nabla H(z_{k+1}), z_{k+1} - z_k \rangle - \frac{L_2}{2}\|z_{k+1} - z_k\|_2^2$$
$$= -\eta\langle \nabla H(z_{k+1}), J\nabla H(z_{k+1})\rangle - \frac{\eta^2 L_2}{2}\|J\nabla H(z_{k+1})\|_2^2$$
$$= 0 - \frac{\eta^2 L_2}{2}\langle \nabla H(z_{k+1}), (J^\top J)\nabla H(z_{k+1})\rangle$$
$$\geq -\frac{\eta^2\alpha_{\max}^2 L_2}{2}\|\nabla H(z_{k+1})\|_2^2$$
$$\geq -\eta^2\alpha_{\max}^2 L_2^2(H(z_{k+1}) - H(z^*))$$

This implies

$$H(z_{k+1}) - H(z^*) \geq \frac{H(z_k) - H(z^*)}{1 + \eta^2 \alpha_{\max}^2 L_2^2}.$$

Therefore, along the backward method, the energy function decreases at most exponentially fast:

$$H(z_k) - H(z^*) \geq \frac{H(z_0) - H(z^*)}{(1 + \eta^2 \alpha_{\max}^2 L_2^2)^k}.$$

$\square$

In Section C.4 we analyze the total regret of the backward method for the min-max game application, and show the duality gap of the average iterate decays at an $O(K^{-1})$ rate; see Theorem C.3. This is similar to the performance of the clairvoyant multiplicative weight update [25] which achieves constant regret for a general game.

### B.3 Alternating Discretization of Skew-Gradient Flow

#### B.3.1 Proof of Corollary 4.3

*Proof of Corollary 4.3.* From Lemma 4.2 and since the Bregman commutator of $H$ vanishes,

$$H_\eta(z_k) = H_\eta(z_0) + \sum_{i=0}^{k-1} C_H(z_{i+1}, z_i) = \tilde{H}_\eta(z_0).$$

$\square$

#### B.3.2 Proof of Lemma 4.2

*Proof of Lemma 4.2.* By the definition of Bregman commutator, and using the alternating update, we can write

$$
\begin{aligned}
C_f(x_{k+1}, x_k) &= f(x_{k+1}) - f(x_k) - \frac{1}{2}\langle \nabla f(x_k) + \nabla f(x_{k+1}), x_{k+1} - x_k \rangle \\
&= f(x_{k+1}) - f(x_k) + \frac{\eta}{2}\langle \nabla f(x_k) + \nabla f(x_{k+1}), A\nabla g(y_k) \rangle.
\end{aligned}
$$

Similarly, we can also write

$$
\begin{aligned}
C_g(y_{k+1}, y_k) &= g(y_{k+1}) - g(y_k) - \frac{1}{2}\langle \nabla g(y_k) + \nabla g(y_{k+1}), y_{k+1} - y_k \rangle \\
&= g(y_{k+1}) - g(y_k) - \frac{\eta}{2}\langle \nabla g(y_k) + \nabla g(y_{k+1}), A^\top \nabla f(x_{k+1}) \rangle \\
&= g(y_{k+1}) - g(y_k) - \frac{\eta}{2}\langle \nabla f(x_{k+1}), A(\nabla g(y_k) + \nabla g(y_{k+1})) \rangle.
\end{aligned}
$$

Since $H(x, y) = f(x) + g(y)$ is separable, we have $D_H(z', z) = D_f(x', x) + D_g(y', y)$ where $z' = (x', y')$ and $z = (x, y)$, and thus $C_H(z', z) = C_f(x', x) + C_g(y', y)$. Adding the two identities above yields

$$
\begin{aligned}
C_H(z_{k+1}, z_k) &= C_f(x_{k+1}, x_k) + C_g(y_{k+1}, y_k) \\
&= f(x_{k+1}) - f(x_k) + \frac{\eta}{2}\langle \nabla f(x_k) + \nabla f(x_{k+1}), A\nabla g(y_k) \rangle \\
&\quad + g(y_{k+1}) - g(y_k) - \frac{\eta}{2}\langle \nabla f(x_{k+1}), A(\nabla g(y_k) + \nabla g(y_{k+1})) \rangle \\
&= H(z_{k+1}) - H(z_k) + \frac{\eta}{2}\langle \nabla f(x_k), A\nabla g(y_k) \rangle - \frac{\eta}{2}\langle \nabla f(x_{k+1}), A\nabla g(y_{k+1}) \rangle \\
&= H_\eta(z_{k+1}) - H_\eta(z_k)
\end{aligned}
$$

as desired.

$\square$

### B.3.3 Proof of Theorem 4.4

*Proof of Theorem 4.4.* From Lemma 4.2 and using Lemma A.2,

$$|H_\eta(z_k) - H_\eta(z_0)| = \left| \sum_{i=0}^{k-1} C_H(z_{i+1}, z_i) \right| \leq \sum_{i=0}^{k-1} |C_H(z_{i+1}, z_i)| \leq \frac{L_3}{12} \sum_{i=0}^{k-1} \|z_{i+1} - z_i\|^3.$$

Along the alternating method (11), we have

$$z_{i+1} - z_i = \eta \begin{pmatrix} -A\nabla g(y_i) \\ A^\top \nabla f(x_{i+1}) \end{pmatrix} = -\eta J \nabla H(z_{i+\frac{1}{2}})$$

where $z_{i+\frac{1}{2}} := \begin{pmatrix} x_{i+1} \\ y_i \end{pmatrix}$. Thus,

$$\|z_{i+1} - z_i\| = \eta \|J\nabla H(z_{i+\frac{1}{2}})\| \leq \eta\, \alpha_{\max} L_1.$$

Therefore,

$$|H_\eta(z_k) - H_\eta(z_0)| \leq \frac{L_3}{12} \sum_{i=0}^{k-1} (\eta\, \alpha_{\max} L_1)^3 = \frac{1}{12} \eta^3\, \alpha_{\max}^3\, L_1^3\, L_3\, k$$

as desired. $\qquad\square$

### B.3.4 Proof of Theorem 4.5

*Proof of Theorem 4.5.* Recall from (6) the regret of the first player is

$$R_{1,K}(p) = \sum_{k=0}^{K-1} \left( \frac{p_k + p_{k+1}}{2} \right)^\top Aq_k - \sum_{k=0}^{K-1} p^\top Aq_k.$$

Recall also $p_k = \nabla f(x_k)$, $q_k = \nabla g(y_k)$, $p = \nabla f(x)$, and $q = \nabla g(y)$.

From the alternating mirror descent update (3), we have

$$Aq_k = A\nabla g(y_k) = -\frac{1}{\eta}(x_{k+1} - x_k).$$

Therefore, the second term in the first player's regret is

$$\sum_{k=0}^{K-1} p^\top Aq_k = -\frac{1}{\eta} \sum_{k=0}^{K-1} \nabla f(x)^\top (x_{k+1} - x_k) = -\frac{1}{\eta} \nabla f(x)^\top (x_K - x_0).$$

For the first term in the first player's regret, by the definition of the Bregman commutator, we have

$$\begin{aligned}
\sum_{k=0}^{K-1} \left( \frac{p_k + p_{k+1}}{2} \right)^\top Aq_k &= -\frac{1}{\eta} \sum_{k=0}^{K-1} \left( \frac{\nabla f(x_{k+1}) + \nabla f(x_k)}{2} \right)^\top (x_{k+1} - x_k) \\
&= -\frac{1}{\eta} \sum_{k=0}^{K-1} \left( f(x_{k+1}) - f(x_k) - C_f(x_{k+1}, x_k) \right) \\
&= -\frac{1}{\eta} \left( f(x_K) - f(x_0) \right) + \frac{1}{\eta} \sum_{k=0}^{K-1} C_f(x_{k+1}, x_k).
\end{aligned}$$

Combining the two terms above, we can write the regret of the first player as

$$\begin{aligned}
R_{1,K}(p) &= -\frac{1}{\eta} \left( f(x_K) - f(x_0) - \nabla f(x)^\top (x_K - x_0) \right) + \frac{1}{\eta} \sum_{k=0}^{K-1} C_f(x_{k+1}, x_k) \\
&= -\frac{1}{\eta} \left( D_f(x_K, x) - D_f(x_0, x) \right) + \frac{1}{\eta} \sum_{k=0}^{K-1} C_f(x_{k+1}, x_k).
\end{aligned} \tag{19}$$

Similarly, recall from (7) the regret of the second player is

$$R_{2,K}(q) = \sum_{k=0}^{K-1} p_{k+1}^\top Aq - \sum_{k=0}^{K-1} p_{k+1}^\top A\left(\frac{q_k + q_{k+1}}{2}\right).$$

From the alternating mirror descent update (3), we have

$$A^\top p_{k+1} = A^\top \nabla f(x_{k+1}) = \frac{1}{\eta}(y_{k+1} - y_k).$$

Therefore, the first term in the second player's regret is

$$\sum_{k=0}^{K-1} p_{k+1}^\top Aq = \frac{1}{\eta}\sum_{k=0}^{K-1}(y_{k+1} - y_k)^\top \nabla g(y) = \frac{1}{\eta}(y_K - y_0)^\top \nabla g(y).$$

For the second term in the second player's regret, by the definition of the Bregman commutator, we have

$$\sum_{k=0}^{K-1} p_{k+1}^\top A\left(\frac{q_k + q_{k+1}}{2}\right) = \frac{1}{\eta}\sum_{k=0}^{K-1}(y_{k+1} - y_k)^\top \left(\frac{\nabla g(y_k) + \nabla g(y_{k+1})}{2}\right)$$

$$= \frac{1}{\eta}\sum_{k=0}^{K-1}\left(g(y_{k+1}) - g(y_k) - C_g(y_{k+1}, y_k)\right)$$

$$= \frac{1}{\eta}(g(y_K) - g(y_0)) - \frac{1}{\eta}\sum_{k=0}^{K-1} C_g(y_{k+1}, y_k).$$

Combining the two terms above, we can write the regret of the second player as

$$R_{2,K}(q) = \frac{1}{\eta}\left(-g(y_K) + g(y_0) + (y_K - y_0)^\top \nabla g(y)\right) + \frac{1}{\eta}\sum_{k=0}^{K-1} C_g(y_{k+1}, y_k)$$

$$= -\frac{1}{\eta}\left(D_g(y_K, y) - D_g(y_0, y)\right) + \frac{1}{\eta}\sum_{k=0}^{K-1} C_g(y_{k+1}, y_k). \tag{20}$$

Adding (19) and (20), we find that the cumulative regret of both players at iteration $K$ is

$$R_K(p, q) = R_{1,K}(p) + R_{2,K}(q)$$

$$= -\frac{1}{\eta}\left(D_f(x_K, x) - D_f(x_0, x)\right) + \frac{1}{\eta}\sum_{k=0}^{K-1} C_f(x_{k+1}, x_k)$$

$$\quad - \frac{1}{\eta}\left(D_g(y_K, y) - D_g(y_0, y)\right) + \frac{1}{\eta}\sum_{k=0}^{K-1} C_g(y_{k+1}, y_k)$$

$$= -\frac{1}{\eta}\left(D_H(z_K, z) - D_H(z_0, z)\right) + \frac{1}{\eta}\sum_{k=0}^{K-1} C_H(z_{k+1}, z_k)$$

$$\overset{(13)}{=} -\frac{1}{\eta}\left(D_H(z_K, z) - D_H(z_0, z)\right) + \frac{1}{\eta}\sum_{k=0}^{K-1}\left(H_\eta(z_{k+1}) - H_\eta(z_k)\right)$$

$$= -\frac{1}{\eta}\left(D_H(z_K, z) - D_H(z_0, z)\right) + \frac{1}{\eta}\left(H_\eta(z_K) - H_\eta(z_0)\right).$$

In the penultimate line above we have applied (13) from Lemma 4.2 to express the Bregman commutator as a difference of the modified energy. □

# C  Details for Regret Calculations

## C.1  Regret Proofs for Alternating Mirror Descent

### C.1.1  Proof of Lemma 3.1

*Proof of Lemma 3.1.* By the definition of duality gap and total regret of the alternating method,

$$
\begin{aligned}
K \cdot \mathsf{dg}(\bar{p}_K, \bar{q}_K) &= K \cdot \max_{(p,q) \in \mathcal{P} \times \mathcal{Q}} \left( \bar{p}_K^\top A q - p^\top A \bar{q}_K \right) \\
&= \max_{(p,q) \in \mathcal{P} \times \mathcal{Q}} \left( \sum_{k=0}^{K-1} p_{k+1}^\top A q - \sum_{k=0}^{K-1} p^\top A q_k \right) \\
&= \max_{(p,q) \in \mathcal{P} \times \mathcal{Q}} \left( R_{1,K}(p) + R_{2,K}(q) - \frac{1}{2}(p_0^\top A q_0 - p_K^\top A q_K) \right) \\
&= R_K - \frac{1}{2}(p_0^\top A q_0 - p_K^\top A q_K).
\end{aligned}
$$

$\square$

### C.1.2  Proof of Theorem 3.2

*Proof of Theorem 3.2.* We use the relation between cumulative regret of alternating mirror descent and the modified energy from the skew-gradient flow discretization in Theorem 4.5.

For any reference point $(p,q) \in \mathcal{P} \times \mathcal{Q}$, let $z = (x,y) \in \mathbb{R}^{m+n}$ be the dual variables, where $x = \nabla\phi(p)$ and $y = \nabla\psi(q)$. Recall we define the energy function $H(x,y) = f(x) + g(y)$ where $f = \phi^*$ and $g = \psi^*$. Along the alternating mirror descent (1), let $(x_k, y_k)$ be the dual variables to $(p_k, q_k)$. By Theorem 4.5, we have

$$
\begin{aligned}
R_K(p,q) &= \frac{D_H(z_0, z) - D_H(z_K, z)}{\eta} + \frac{H_\eta(z_K) - H_\eta(z_0)}{\eta} \\
&\leq \frac{D_H(z_0, z)}{\eta} + \frac{H_\eta(z_K) - H_\eta(z_0)}{\eta}.
\end{aligned}
\tag{21}
$$

The inequality in the last line above follows from $D_H(z_K, z) \geq 0$ since $H$ is convex.

We verify that the assumptions in Theorem 3.2 imply the assumptions in Theorem 4.4 are satisfied. Indeed, $H$ is $2M$-Lipschitz since its gradient is $\nabla H(x,y) = (\nabla f(x), \nabla g(y)) = (p,q)$, so

$$
\|\nabla H(x,y)\| = \|(p,q)\| \leq \|p\| + \|q\| \leq M + M = 2M.
$$

Furthermore, $H$ is $2M$-smooth of order 3 since

$$
\nabla^3 H(x,y) = (\nabla^3 f(x), \nabla^3 g(y)) = (\nabla^3 \phi^*(\nabla\phi(p)), \nabla^3 \psi^*(\nabla\psi(q)))
$$

so $\|\nabla^3 H(x,y)\|_{\mathrm{op}} \leq \|\nabla^3 \phi^*(\nabla\phi(p))\|_{\mathrm{op}} + \|\nabla^3 \psi^*(\nabla\psi(q))\|_{\mathrm{op}} \leq M + M = 2M$.

Then by Theorem 4.4,

$$
|H_\eta(z_K) - H_\eta(z_0)| \leq \frac{\alpha_{\max}^3 (2M)(2M)^3}{12} \eta^3 K = \frac{4}{3} \alpha_{\max}^3 M^4 \eta^3 K.
$$

Plugging this to (21) gives

$$
R_K(p,q) \leq \frac{D_H(z_0, z)}{\eta} + \frac{4}{3} \alpha_{\max}^3 M^4 \eta^2 K.
$$

Since

$$
D_H(z_0, z) = D_f(x_0, x) + D_g(y_0, y) = D_\phi(p, p_0) + D_\psi(q, q_0)
$$

we also have

$$
R_K(p,q) \leq \frac{D_\phi(p, p_0) + D_\psi(q, q_0)}{\eta} + \frac{4}{3} \alpha_{\max}^3 M^4 \eta^2 K.
\tag{22}
$$

Now suppose we start from $(p_0, q_0)$ such that $\sup_{p \in \mathcal{P}} D_\phi(p, p_0) \le M_2$ and $\sup_{q \in \mathcal{Q}} D_\psi(q, q_0) \le M_2$ for some $0 < M_2 < \infty$. Then we can maximize (22) over $(p, q) \in \mathcal{P} \times Q$ to get the total regret

$$R_K = \sup_{(p,q) \in \mathcal{P} \times Q} R_K(p, q) \le \frac{2M_2}{\eta} + \frac{4}{3} \alpha_{\max}^3 M^4 \eta^2 K.$$

With the choice of step size $\eta = \Theta(K^{-1/3})$, we obtain the bound $R_K = O(K^{1/3})$. $\qquad \square$

### C.1.3 Proof of Corollary 3.3

*Proof of Corollary 3.3.* By assumption and Theorem 3.2, we have $R_K = O(K^{1/3})$, so $\frac{1}{K} R_K = O(K^{-2/3})$. Since we assume the domains are bounded, we have

$$|p_0^\top A q_0| \le \alpha_{\max} \|p_0\| \cdot \|q_0\| \le \alpha_{\max} M^2$$

and similarly, $|p_K^\top A q_K| \le \alpha_{\max} M^2$. Thus, the last term in (8) is bounded by

$$\frac{1}{K} |p_0^\top A q_0 - p_K^\top A q_K| \le \frac{2\alpha_{\max} M^2}{K} = O(K^{-1}).$$

Therefore, by Lemma 3.1, $\mathrm{dg}(\bar{p}_K, \bar{q}_K) = O(K^{-2/3}) + O(K^{-1}) = O(K^{-2/3})$. $\qquad \square$

### C.2 Regret of the Continuous-time Method

Suppose the two players follow the continuous-time variant of mirror descent, namely the natural gradient flow:

$$\dot{P}_t = -\nabla^2 \phi(P_t)^{-1} A Q_t$$
$$\dot{Q}_t = \nabla^2 \psi(Q_t)^{-1} A^\top P_t.$$

This is the continuous-time limit (as the step size $\eta \to 0$) of the mirror descent method—either the simultaneous, alternating, or proximal version. In terms of the dual variables $X_t = \nabla \phi(P_t)$, $Y_t = \nabla \psi(Q_t)$, they follow the skew-gradient flow (4):

$$\dot{X}_t = -A \nabla g(Y_t)$$
$$\dot{Y}_t = A^\top \nabla f(X_t)$$

where $f = \phi^*$ and $g = \psi^*$. Recall in this case the energy function $H(x, y) = f(x) + g(y)$ is conserved: $H(X_t, Y_t) = H(X_0, Y_0)$, which means the trajectories $(X_t, Y_t)$ cycle and stay in the orbit of constant energy.

For the min-max game application, we show this continuous-time method achieves constant regret.

We define the regret of the first player at (continuous) time $T$ with respect to $\hat{p} \in \mathcal{P}$ to be:

$$R_{1,T}(\hat{p}) := \int_0^T P_t^\top A Q_t \, dt - \int_0^T \hat{p}^\top A Q_t \, dt.$$

Similarly, we define the regret of the second player at time $T$ with respect to $\hat{q} \in \mathcal{Q}$ to be:

$$R_{2,T}(\hat{q}) := \int_0^T P_t^\top A \hat{q} \, dt - \int_0^T P_t^\top A Q_t \, dt.$$

We define the cumulative regret of both players at time $T$ with respect to $(\hat{p}, \hat{q}) \in \mathcal{P} \times \mathcal{Q}$ to be:

$$R_T(\hat{p}, \hat{q}) := R_{1,T}(\hat{p}) + R_{2,T}(\hat{q}) = \int_0^T P_t^\top A \hat{q} \, dt - \int_0^T \hat{p}^\top A Q_t \, dt.$$

Finally, we define the **total regret** at time $T$ to be the maximum cumulative regret of both players:

$$R_T := \max_{(\hat{p}, \hat{q}) \in \mathcal{P} \times \mathcal{Q}} R_T(\hat{p}, \hat{q}).$$

Define the average iterates of the two players at time $T$:

$$\bar{P}_T = \frac{1}{T} \int_0^T P_t \, dt \qquad \text{and} \qquad \bar{Q}_T = \frac{1}{T} \int_0^T Q_t \, dt$$

We observe that the average total regret is equal to the duality gap of the average iterates:

$$\frac{1}{T} R_T = \max_{q \in \mathcal{Q}} \bar{P}_T^\top A q - \min_{p \in \mathcal{P}} p^\top A \bar{Q}_T = \mathsf{dg}(\bar{p}_T, \bar{q}_T).$$

Then we have the following properties.

**Theorem C.1.** *Assume the domains $\mathcal{P}$ and $\mathcal{Q}$ are bounded, so there exists $0 < M < \infty$ such that $D_\phi(p', p) \le M$ and $D_\psi(q', q) \le M$ for all $p', p \in \mathcal{P}$, $q', q \in \mathcal{Q}$. If both players play the natural gradient flow dynamics (23), then the total regret is bounded for all $T \ge 0$:*

$$R_T \le 2M.$$

*Therefore, the duality gap of the average iterate decays at an $O(1/T)$ rate:*

$$\mathsf{dg}(\bar{P}_T, \bar{Q}_T) \le \frac{2M}{T}.$$

*Proof.* Let $\hat{x} = \nabla \phi(\hat{p})$ and $\hat{y} = \nabla \psi(\hat{q})$, so $\hat{p} = \nabla f(\hat{x})$ and $\hat{q} = \nabla g(\hat{y})$. By the definition of the dynamics, we have

$$
\begin{aligned}
(P_t - \hat{p})^\top A Q_t &= -(\nabla f(X_t) - \nabla f(\hat{x}))^\top \dot{X}_t \\
&= \frac{d}{dt}(-f(X_t) + \langle \nabla f(\hat{x}), X_t - \hat{x} \rangle) \\
&= -\frac{d}{dt} D_f(X_t, \hat{x}).
\end{aligned}
$$

Therefore, we can write the regret of the first player as

$$
\begin{aligned}
R_{1,T}(\hat{p}) &= \int_0^T (P_t - \hat{p})^\top A Q_t \, dt \\
&= D_f(X_0, \hat{x}) - D_f(X_T, \hat{x}) \\
&= D_\phi(\hat{p}, P_0) - D_\phi(\hat{p}, P_T).
\end{aligned}
$$

Similarly, by the definition of the dynamics, we have

$$
\begin{aligned}
P_t^\top A(\hat{q} - Q_t) &= \dot{Y}_t^\top (\nabla g(\hat{y}) - \nabla g(Y_t))^\top \\
&= \frac{d}{dt}(-g(Y_t) + \langle \nabla g(\hat{y}), Y_t - \hat{y} \rangle) \\
&= -\frac{d}{dt} D_g(Y_t, \hat{y}).
\end{aligned}
$$

Therefore, we can write the regret of the second player as

$$
\begin{aligned}
R_{2,T}(\hat{q}) &= \int_0^T P_t^\top A(\hat{q} - Q_t) \, dt \\
&= D_g(Y_0, \hat{y}) - D_g(Y_T, \hat{y}) \\
&= D_\psi(\hat{q}, Q_0) - D_\psi(\hat{q}, Q_T).
\end{aligned}
$$

Combining the two quantities above, and since $\phi, \psi$ are convex, we find that

$$
\begin{aligned}
R_T(\hat{p}, \hat{q}) &= D_\phi(\hat{p}, P_0) - D_\phi(\hat{p}, P_T) + D_\psi(\hat{q}, Q_0) - D_\psi(\hat{q}, Q_T) \\
&\le D_\phi(\hat{p}, P_0) + D_\psi(\hat{q}, Q_0) \\
&\le 2M.
\end{aligned}
$$

Therefore the total regret is also bounded:

$$R_T = \max_{(\hat{p}, \hat{q}) \in \mathcal{P} \times \mathcal{Q}} R_T(\hat{p}, \hat{q}) \le 2M.$$

Thus, the duality gap of the average iterate is bounded by:

$$\mathsf{dg}(\bar{P}_T, \bar{Q}_T) = \frac{1}{T} R_T \le \frac{2M}{T}.$$

$\square$

### C.3 Regret of the Forward Method

Consider the forward method (17) for discretizing the skew-gradient flow (4), which means the two players follow the simultaneous mirror descent algorithm. Recall from Theorem B.1 that the energy function is increasing along the forward method. We show that the total regret of the two players increases at an $O(K^{1/2})$ rate, and thus the average total regret decays as $O(K^{-1/2})$ after $K$ iterations, which recovers the classical rate of simultaneous mirror descent.

We define the regret of the first player at iteration $K$ with respect to $\hat{p} \in \mathcal{P}$ to be:

$$R_{1,K}(\hat{p}) := \sum_{k=0}^{K-1} p_k^\top A q_k - \sum_{k=0}^{K-1} \hat{p}^\top A q_k.$$

Similarly, we define the regret of the second player at iteration $K$ with respect to $\hat{q} \in \mathcal{Q}$ to be:

$$R_{2,K}(\hat{q}) := \sum_{k=0}^{K-1} p_k^\top A \hat{q} - \sum_{k=0}^{K-1} p_k^\top A q_k.$$

We define the cumulative regret of both players at iteration $K$ with respect to $(\hat{p}, \hat{q}) \in \mathcal{P} \times \mathcal{Q}$ to be:

$$R_K(\hat{p}, \hat{q}) := R_{1,K}(\hat{p}) + R_{2,K}(\hat{q}) = \sum_{k=0}^{K-1} p_k^\top A \hat{q} - \sum_{k=0}^{K-1} \hat{p}^\top A q_k.$$

Finally, we define the **total regret** at iteration $K$ to be the maximum cumulative regret of both players:

$$R_K := \max_{(\hat{p}, \hat{q}) \in \mathcal{P} \times \mathcal{Q}} R_K(\hat{p}, \hat{q}).$$

Define the average iterates of the two players at iteration $K$:

$$\bar{p}_K = \frac{1}{K} \sum_{k=0}^{K-1} p_k \qquad \text{and} \qquad \bar{q}_K = \frac{1}{K} \sum_{k=0}^{K-1} q_k.$$

We observe that the average regret is equal to the duality gap of the average iterates:

$$\frac{1}{K} R_K = \max_{q \in \mathcal{Q}} \bar{p}_K^\top A q - \min_{p \in \mathcal{P}} p^\top A \bar{q}_K = \mathsf{dg}(\bar{p}_K, \bar{q}_K).$$

Then we have the following properties.

**Theorem C.2.** *Assume the domains $\mathcal{P}$ and $\mathcal{Q}$ are bounded, so there exists $0 < M < \infty$ such that $D_\phi(p', p) \leq M$ and $D_\psi(q', q) \leq M$ for all $p', p \in \mathcal{P}$, $q', q \in \mathcal{Q}$. Assume the energy function $H = f + g = \phi^* + \psi^*$ is $L_1$-Lipschitz and $L_2$-smooth. Let $\alpha_{\max}$ be the largest singular value of $A$. If both players play the forward method (17), then the total regret is bounded by:*

$$R_K \leq \frac{1}{2} \eta \, \alpha_{\max}^2 \, L_1^2 \, L_2 \, K + \frac{2M}{\eta}.$$

*In particular, choosing $\eta = \Theta(K^{-1/2})$ yields the bound $R_K \leq O(K^{1/2})$, and therefore,*

$$\mathsf{dg}(\bar{p}_K, \bar{q}_K) \leq O(K^{-1/2}).$$

*Proof.* Let $\hat{x} = \nabla \phi(\hat{p})$ and $\hat{y} = \nabla \psi(\hat{q})$, so $\hat{p} = \nabla f(\hat{x})$ and $\hat{q} = \nabla g(\hat{y})$. From the forward method update (17),

$$(p_k - \hat{p})^\top A q_k = -\frac{1}{\eta} (\nabla f(x_k) - \nabla f(\hat{x}))^\top (x_{k+1} - x_k)$$

$$= \frac{1}{\eta} (D_f(x_{k+1}, x_k) + D_f(x_k, \hat{x}) - D_f(x_{k+1}, \hat{x}))$$

$$= \frac{1}{\eta} (D_\phi(p_k, p_{k+1}) + D_\phi(\hat{p}, p_k) - D_\phi(\hat{p}, p_{k+1})).$$

Therefore,

$$R_{1,K}(\hat{p}) = \frac{1}{\eta} \sum_{k=0}^{K-1} D_\phi(p_k, p_{k+1}) + \frac{1}{\eta}(D_\phi(\hat{p}, p_0) - D_\phi(\hat{p}, p_K))$$

$$\leq \frac{1}{\eta} \sum_{k=0}^{K-1} D_\phi(p_k, p_{k+1}) + \frac{1}{\eta} D_\phi(\hat{p}, p_0)$$

where the last inequality follows since $\phi$ is convex.

Similarly, from the forward method update (17),

$$p_k^\top A(\hat{q} - q_k) = \frac{1}{\eta}(y_{k+1} - y_k)^\top (\nabla g(\hat{y}) - \nabla g(y_k))$$

$$= \frac{1}{\eta}(D_g(y_{k+1}, y_k) + D_g(y_k, \hat{y}) - D_g(y_{k+1}, \hat{y}))$$

$$= \frac{1}{\eta}(D_\psi(q_k, q_{k+1}) + D_\psi(\hat{q}, q_k) - D_\psi(\hat{q}, q_{k+1})).$$

Therefore,

$$R_{2,K}(\hat{q}) = \frac{1}{\eta} \sum_{k=0}^{K-1} D_\psi(q_k, q_{k+1}) + \frac{1}{\eta}(D_\psi(\hat{q}, q_0) - D_\psi(\hat{q}, q_K))$$

$$\leq \frac{1}{\eta} \sum_{k=0}^{K-1} D_\psi(q_k, q_{k+1}) + \frac{1}{\eta} D_\psi(\hat{q}, q_0)$$

where the last inequality follows since $\psi$ is convex.

Combining the two quantities above, we have

$$R_K(\hat{p}, \hat{q}) \leq \frac{1}{\eta} \sum_{k=0}^{K-1} (D_\phi(p_k, p_{k+1}) + D_\psi(q_k, q_{k+1})) + \frac{1}{\eta}(D_\phi(\hat{p}, p_0) + D_\psi(\hat{q}, q_0))$$

$$= \frac{1}{\eta} \sum_{k=0}^{K-1} D_H(z_{k+1}, z_k) + \frac{1}{\eta} D_H(z_0, \hat{z}).$$

Since $z_{k+1} = z_k - \eta J \nabla H(z_k)$, and $H$ is $L_1$-Lipschitz and $L_2$-smooth, we can bound:

$$D_H(z_{k+1}, z_k) \leq \frac{L_2}{2} \|z_{k+1} - z_k\|^2$$

$$= \frac{\eta^2 L_2}{2} \|J \nabla H(z_k)\|^2$$

$$\leq \frac{\eta^2 \alpha_{\max}^2 L_2}{2} \|\nabla H(z_k)\|^2$$

$$\leq \frac{\eta^2 \alpha_{\max}^2 L_2 L_1^2}{2}.$$

This gives a regret bound:

$$R_K(\hat{p}, \hat{q}) \leq \frac{\eta K \alpha_{\max}^2 L_2 L_1^2}{2} + \frac{1}{\eta} D_H(z_0, \hat{z})$$

$$\leq \frac{\eta K \alpha_{\max}^2 L_2 L_1^2}{2} + \frac{2M}{\eta}$$

where the last inequality follows since $D_H(z_0, \hat{z}) = D_f(x_0, \hat{x}) + D_g(y_0, \hat{y}) \leq 2M$. Thus, the total regret is also bounded:

$$R_K \leq \frac{\eta K \alpha_{\max}^2 L_2 L_1^2}{2} + \frac{2M}{\eta}.$$

Choosing $\eta = \frac{2\sqrt{M}}{\alpha_{\max}L_1\sqrt{L_2 K}} = \Theta(K^{-1/2})$ gives

$$R_K \leq 2\alpha_{\max}L_1\sqrt{L_2 M K} = O(K^{1/2})$$

Therefore, the duality gap of the average iterates is bounded by $\mathsf{dg}(\bar{p}_K, \bar{q}_K) = \frac{1}{K}R_K \leq O(K^{-1/2})$, as desired. □

### C.4 Regret of the Backward Method

Consider the backward method (18) for discretizing the skew-gradient flow (4), which means the two players follow the simultaneous proximal mirror descent algorithm. Recall from Theorem B.3 that the energy function is decreasing along the forward method. We show that the total regret of the two players along the backward method is bounded by a constant, and thus the average total regret decays as $O(K^{-1})$ after $K$ iterations. This is similar to the property of the clairvoyant multiplicative weight update [25], which achieves a constant regret bound for general games.

We define the regret of the first player at iteration $K$ with respect to $\hat{p} \in \mathcal{P}$ to be:

$$R_{1,K}(\hat{p}) := \sum_{k=0}^{K-1} p_{k+1}^\top A q_{k+1} - \sum_{k=0}^{K-1} \hat{p}^\top A q_{k+1}.$$

Note the difference with the forward method; here we are shifting the time index by 1. Similarly, We define the regret of the second player ($q$) at the beginning of iteration $K$ (at time $T = \eta K$) with respect to a static point $\hat{q} \in \mathcal{Q}$ to be:

$$R_{2,K}(\hat{q}) := \sum_{k=0}^{K-1} p_{k+1}^\top A \hat{q} - \sum_{k=0}^{K-1} p_{k+1}^\top A q_{k+1}.$$

We define the cumulative regret of both players at the iteration $K$ with respect to $(\hat{p}, \hat{q}) \in \mathcal{P} \times \mathcal{Q}$ to be:

$$R_K(\hat{p}, \hat{q}) := R_{1,K}(\hat{p}) + R_{2,K}(\hat{q}) = \sum_{k=0}^{K-1} p_{k+1}^\top A \hat{q} - \sum_{k=0}^{K-1} \hat{p}^\top A q_{k+1}.$$

Finally, we define the **total regret** at iteration $K$ to be the maximum cumulative regret of both players:

$$R_K := \max_{(\hat{p},\hat{q}) \in \mathcal{P} \times \mathcal{Q}} R_K(\hat{p}, \hat{q}).$$

Define the average iterates of the two players at iteration $K$:

$$\bar{p}_K = \frac{1}{K}\sum_{k=0}^{K-1} p_{k+1} \qquad \text{and} \qquad \bar{q}_K = \frac{1}{K}\sum_{k=0}^{K-1} q_{k+1}.$$

We observe that the average regret is equal to the duality gap of the average iterates:

$$\frac{1}{K}R_K = \max_{q \in \mathcal{Q}} \bar{p}_K^\top A q - \min_{p \in \mathcal{P}} p^\top A \bar{q}_K = \mathsf{dg}(\bar{p}_K, \bar{q}_K).$$

Then we have the following properties.

**Theorem C.3.** *Assume the domains $\mathcal{P}$ and $\mathcal{Q}$ are bounded, so there exists $0 < M < \infty$ such that $D_\phi(p', p) \leq M$ and $D_\psi(q', q) \leq M$ for all $p', p \in \mathcal{P}$, $q', q \in \mathcal{Q}$. If both players play the backward method (18) for any step size $\eta > 0$, then the total regret is bounded by:*

$$R_K \leq \frac{2M}{\eta}.$$

*In particular, the duality gap decays as:*

$$\mathsf{dg}(\bar{p}_K, \bar{q}_K) \leq \frac{2M}{\eta K}.$$

*Proof.* Let $\hat{x} = \nabla\phi(\hat{p})$ and $\hat{y} = \nabla\psi(\hat{q})$, so $\hat{p} = \nabla f(\hat{x})$ and $\hat{q} = \nabla g(\hat{y})$. From the backward method update (18),

$$
\begin{aligned}
(p_{k+1} - \hat{p})^\top A q_{k+1} &= -\frac{1}{\eta}(\nabla f(x_{k+1}) - \nabla f(\hat{x}))^\top(x_{k+1} - x_k) \\
&= \frac{1}{\eta}(D_f(x_k, \hat{x}) - D_f(x_k, x_{k+1}) - D_f(x_{k+1}, \hat{x})) \\
&= \frac{1}{\eta}(D_\phi(\hat{p}, p_k) - D_\phi(p_{k+1}, p_k) - D_\phi(\hat{p}, p_{k+1})).
\end{aligned}
$$

Therefore,

$$
\begin{aligned}
R_{1,K}(\hat{p}) &= -\frac{1}{\eta}\sum_{k=0}^{K-1} D_\phi(p_k, p_{k+1}) + \frac{1}{\eta}(D_\phi(\hat{p}, p_0) - D_\phi(\hat{p}, p_K)) \\
&\leq \frac{1}{\eta} D_\phi(\hat{p}, p_0)
\end{aligned}
$$

where the last inequality follows since $\phi$ is convex.

Similarly, from the backward method update (18),

$$
\begin{aligned}
p_{k+1}^\top A(\hat{q} - q_{k+1}) &= \frac{1}{\eta}(y_{k+1} - y_k)^\top(\nabla g(\hat{y}) - \nabla g(y_{k+1})) \\
&= \frac{1}{\eta}(D_g(y_k, \hat{y}) - D_g(y_k, y_{k+1}) - D_g(y_{k+1}, \hat{y})) \\
&= \frac{1}{\eta}(D_\psi(\hat{q}, q_k) - D_\psi(q_{k+1}, q_k) - D_\psi(\hat{q}, q_{k+1})).
\end{aligned}
$$

Therefore,

$$
\begin{aligned}
R_{2,K}(\hat{q}) &= -\frac{1}{\eta}\sum_{k=0}^{K-1} D_\psi(q_{k+1}, q_k) + \frac{1}{\eta}(D_\psi(\hat{q}, q_0) - D_\psi(\hat{q}, q_K)) \\
&\leq \frac{1}{\eta} D_\psi(\hat{q}, q_0)
\end{aligned}
$$

where the last inequality follows since $\psi$ is convex.

Combining the two quantities above, we have

$$
\begin{aligned}
R_K(\hat{p}, \hat{q}) &\leq \frac{1}{\eta}(D_\phi(\hat{p}, p_0) + D_\psi(\hat{q}, q_0)) \\
&\leq \frac{2M}{\eta}.
\end{aligned}
$$

Therefore,

$$
\mathsf{dg}(\bar{p}_K, \bar{q}_K) = \frac{1}{K} R_K \leq \frac{2M}{\eta K}.
$$

$\square$

# D  Toward a Better Regret Bound under Hamiltonian Structure

There are still some gaps between our results for alternating mirror descent for constrained min-max games and the results from [2] for alternating gradient descent for unconstrained min-max games. In particular, our bound in Theorem 4.4 still grows with the number of iterations, and does not yield constant regret, unlike in the unconstrained case. We believe that it is possible to close this gap in some particular settings, for example when the sysem has a Hamiltonian structure.

In the special case when $m = n$ and the payoff matrix is the identity matrix $A = I$, the skew-gradient flow dynamics (9) becomes a *Hamiltonian flow*; this means in addition to conserving the energy

function, the continuous dynamics also preserves the canonical symplectic structure. (When the payoff matrix $A$ is arbitrary, the skew-gradient flow dynamics has a non-canonical Hamiltonian structure, for which some of the properties still apply; for simplicity, here we consider identity payoff matrix.) In this case, the update (2) for alternating mirror descent becomes what is known as the *symplectic Euler* discretization method, which also preserves the symplectic structure. From classical results in numerical analysis (e.g. see [12]), the symplectic Euler method has a good energy conservation property in discrete time. In particular, informally, under some nice assumptions on the energy function and the trajectories of the algorithm, the symplectic Euler method approximately preserves the energy function with bounded $O(\eta)$ error for exponentially many (in terms of step size) number of iterations. Furthermore, using the technique of backward error analysis, there is a modified energy function (expressed in terms of a formal series) which the discrete-time algorithm should conserve. When translated into the min-max game application, this would yield a poly-logarithmic bound on the total regret. However, to apply these powerful tools, rather nontrivial conditions need to be satisfied (e.g., integrability, and Diophantine condition). On the other hand, a key property of the energy function that we have in our context, namely convexity, has not been utilized by these tools. Therefore, we leave as a future work the task of quantifying with concrete bounds the performance of the symplectic Euler method. Here we simply conjecture that for sufficiently nice and *convex* energy functions, the iterates of the alternating method with small enough step size stay uniformly bounded, as in the unconstrained setting.

**Conjecture 1.** *Assume H is convex, differentiable, and satisfies some mild assumptions on the growth of derivatives. There exists $\eta_{\max} > 0$ such that for all step sizes $0 < \eta < \eta_{\max}$, along the alternating method* (3)*, the modified energy function remains bounded:*

$$\sup_{k \geq 0} |H_\eta(z_k)| < \infty.$$

As a concrete example, we conjecture this property holds for the *alternating multiplicative weight update algorithm*, which is the alternating mirror descent method for simplex constraints with negative entropy regularizer. In this case, the energy function in the dual space is given by

$$H(x, y) = \log \sum_{i=1}^m e^{x_i} + \log \sum_{j=1}^n e^{y_j}.$$

Below, we provide some empirical evidence supporting this conjecture.

### D.1 Simulation results in one-dimension

We present some simulation results in one-dimension that supports the conjecture above. For simplicity we take $A = 1$ (otherwise we can rescale both $f$ and $g$). We provide the code in Section D.2.

For each example, we plot the trajectory of $(x_k, y_k)$ on the left figure, and we plot the energy function and the modified energy function on the right figure.

### D.1.1 Quadratic

As a sanity check, let $f(x) = g(x) = \frac{1}{2}x^2$. This corresponds to the unconstrained ($\mathcal{P} = \mathcal{Q} = \mathbb{R}$) min-max game with quadratic regularizer, i.e. the alternating gradient descent method of [2].

In Figure 1 we use initial position $z_0 = (x_0, y_0) = (3, 3)$, step size $\eta = 0.1$, and number of iterations $K = 300$. We see the modified energy function is conserved exactly, and the trajectory is very close to a circle.

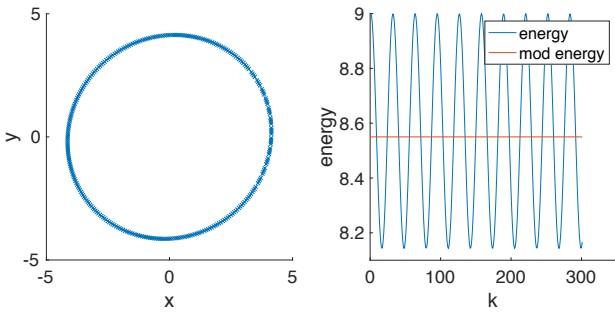

Figure 1: $f$ and $g$ are quadratic with $z_0 = (3,3)$, $\eta = 0.1$, $K = 300$.

In Figure 2 we use $z_0 = (x_0, y_0) = (3,3)$, $\eta = 1.1$, and $K = 50$. The modified energy function is still conserved exactly. The trajectory is not a circle anymore (it is an ellipse), but still bounded.

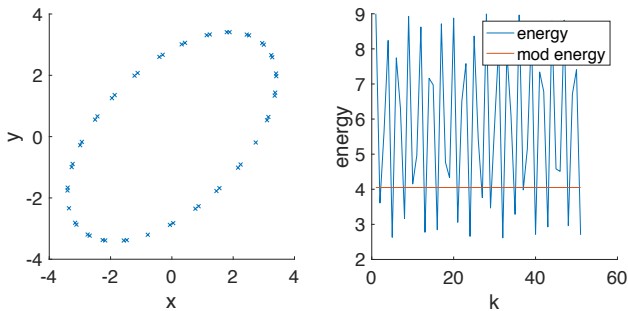

Figure 2: $f$ and $g$ are quadratic with $z_0 = (3,3)$, $\eta = 1.1$, $K = 50$.

### D.1.2  Log-cosh

Let $f(x) = g(x) = \log \cosh(x)$. This corresponds to the min-max game with two-dimensional simplex constraints ($\mathcal{P} = \mathcal{Q} = \Delta_2 \subset \mathbb{R}^2$) with negative entropy regularizers. In the dual space, the dual function becomes $\phi^*(x_1, x_2) = \log(e^{x_1} + e^{x_2}) = \frac{x_1 + x_2}{2} + \log \cosh(\frac{x_1 - x_2}{2}) + \log 2$. So up to a linear function, the dual function becomes $f(x) = \log \cosh x$ with $x = \frac{x_1 - x_2}{2}$.

In Figure 3 we use initial position $z_0 = (x_0, y_0) = (3,3)$, step size $\eta = 0.1$, and number of iterations $K = 300$. Other initial positions give the same qualitative behavior. We see the modified energy function is almost conserved and the trajectory is almost periodic.

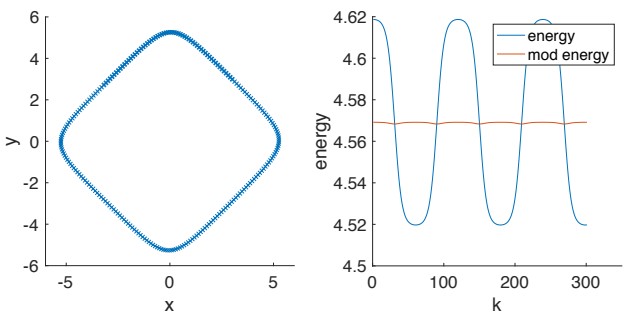

Figure 3: $f$ and $g$ are $\log \cosh$ with $z_0 = (3,3)$, $\eta = 0.1$, $K = 300$.

In Figure 4 we use $z_0 = (x_0, y_0) = (3, 3)$, $\eta = 1$, and $K = 200$.

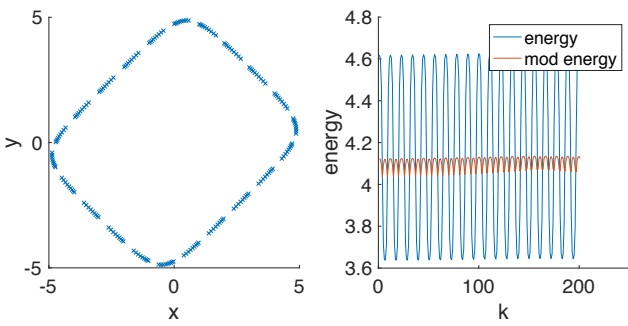

Figure 4: $f$ and $g$ are $\log \cosh$ with $z_0 = (3, 3)$, $\eta = 1$, $K = 200$.

In Figure 5 we use $z_0 = (x_0, y_0) = (3, 3)$, $\eta = 3$, and $K = 800$.

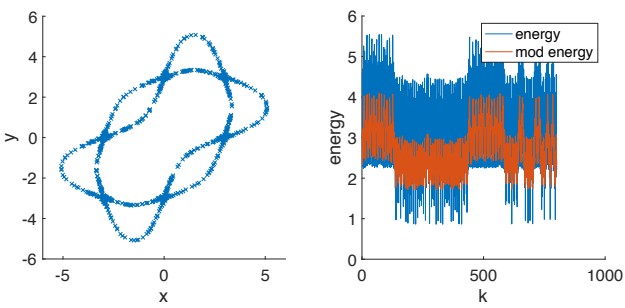

Figure 5: $f$ and $g$ are $\log \cosh$ with $z_0 = (3, 3)$, $\eta = 3$, $K = 800$.

### D.1.3 Quadratic and log cosh

Let $f(x) = \frac{1}{2}x^2$ and $g(x) = \log \cosh(x)$. This corresponds to a min-max game with one-dimensional unconstrained space for the first player with quadratic regularizer, and a two-dimensional simplex constraint for the second player with negative entropy regularizer.

In Figure 6 we use initial position $z_0 = (x_0, y_0) = (3, 3)$, step size $\eta = 0.1$, and number of iterations $K = 300$.

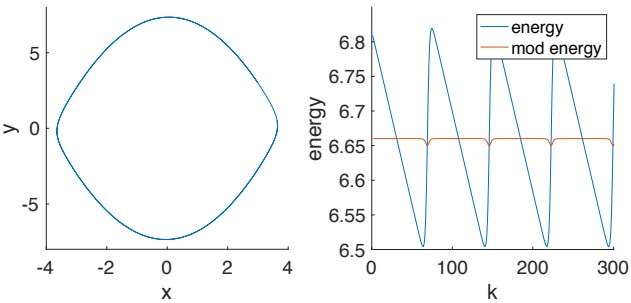

Figure 6: $f$ is quadratic and $g$ is $\log \cosh$ with $z_0 = (3, 3)$, $\eta = 0.1$, $K = 300$.

In Figure 7 we use $z_0 = (x_0, y_0) = (3, 3)$, $\eta = 1$, and $K = 800$.

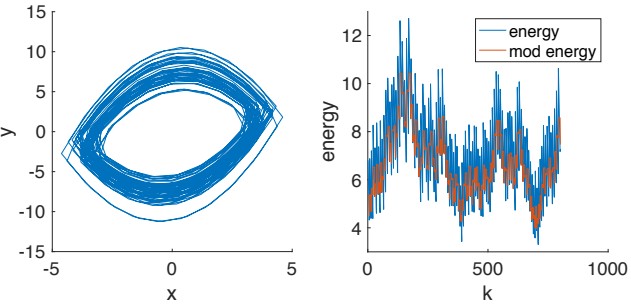

Figure 7: $f$ is quadratic and $g$ is $\log \cosh$ with $z_0 = (3, 3)$, $\eta = 1$, $K = 800$.

### D.1.4 Cubic

Let $f(x) = g(x) = \frac{1}{3}|x|^3$. In Figure 8 we use initial position $z_0 = (x_0, y_0) = (3, 3)$, step size $\eta = 0.1$, and number of iterations $K = 300$.

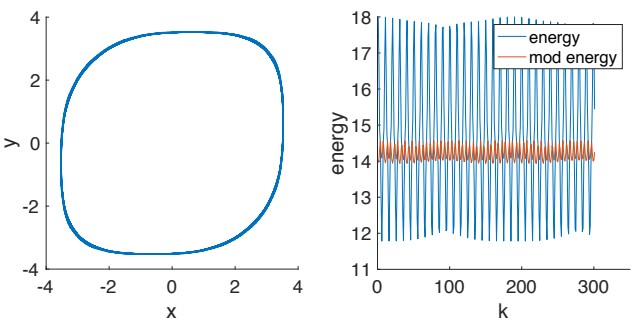

Figure 8: $f$ and $g$ are cubic with $z_0 = (3, 3)$, $\eta = 0.1$, $K = 300$.

In Figure 9 we use $z_0 = (x_0, y_0) = (3, 3)$, $\eta = 0.3$, and $K = 100$.

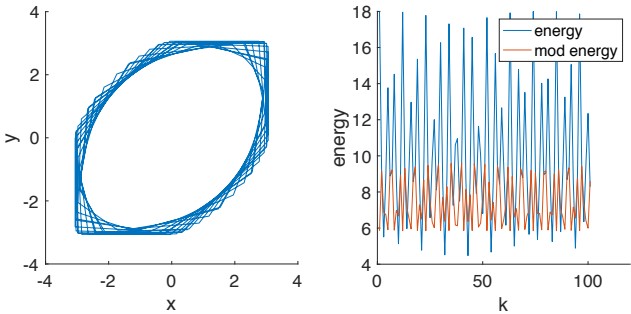

Figure 9: $f$ and $g$ are cubic with $z_0 = (3, 3)$, $\eta = 0.3$, $K = 100$.

### D.2 Code for simulation

This is the MATLAB code for the simulation results presented above. We ran the simulations on a personal computer with 2.7 GHz processor and 16 GB of RAM. Each simulation took less than 2 seconds.

```matlab
%% Script to simulate alternating method

% Define functions and their gradients

% Polynomial of degree p
polyp = @(p,x) 1/p.*abs(x).^p;
grad_polyp = @(p,x) x.*abs(x).^(p-2);

% log-cosh (log-sum-exp in 1-dimension)
logcosh = @(x) log(cosh(x));
grad_logcosh = @(x) tanh(x);

% Choose f and g from the list above, or define explicitly

% First option: Quadratic
f = @(x) polyp(2,x);
gradf = @(x) grad_polyp(2,x);
g = @(y) polyp(2,y);
gradg = @(y) grad_polyp(2,y);

% % Second option: logcosh
% f = logcosh;
% gradf = grad_logcosh;
% g = logcosh;
% gradg = grad_logcosh;

% % Third option: quadratic and logcosh
% f = @(x) polyp(2,x);
% gradf = @(x) grad_polyp(2,x);
% g = logcosh;
% gradg = grad_logcosh;

% % Fourth option: cubic
% f = @(x) polyp(3,x);
% gradf = @(x) grad_polyp(3,x);
% g = @(x) polyp(3,x);
% gradg = @(x) grad_polyp(3,x);

% Initial point
x0 = 3;
y0 = 3;

K = 200;   % number of rounds
eta = 1;   % step size

% Define energy and modified energy
energy = @(x,y) f(x) + g(y);
modenergy = @(x,y) energy(x,y) - eta/2.*gradf(x).*gradg(y);

% Prepare storage
xList = zeros(K+1,1);
yList = zeros(K+1,1);
energyList = zeros(K+1,1);
modenergyList = zeros(K+1,1);
```

```matlab
xList(1) = x0;
yList(1) = y0;
energyList(1) = energy(x0,y0);
modenergyList(1) = modenergy(x0,y0);

% Iterate
for k = 1:K
    curX = xList(k);
    curY = yList(k);
    newX = curX - eta*gradg(curY);
    newY = curY + eta*gradf(newX);
    xList(k+1) = newX;
    yList(k+1) = newY;
    energyList(k+1) = energy(newX,newY);
    modenergyList(k+1) = modenergy(newX,newY);
end

% Plot
fig = figure;

subplot(1,2,1); hold all; set(gca,'FontSize',16);
axis square;
scatter(xList,yList,'x');
xlabel('x');
ylabel('y');

subplot(1,2,2); hold all; set(gca,'FontSize',16);
axis square;
plot(energyList);
plot(modenergyList);
xlabel('k');
ylabel('energy');
legend('energy', 'mod energy');

set(fig,'PaperOrientation','landscape');
set(fig,'PaperUnits','normalized');
set(fig,'PaperPosition', [0 0 1 1]);
```