# OpenReview forum: "Alternating Mirror Descent for Constrained Min-Max Games"
_NeurIPS.cc/2022/Conference — NeurIPS 2022 Accept_

### Official Review · Reviewer_SuUA · 2022-07-14

**Rating:** 6
**Confidence:** 3
**Soundness:** 4 excellent
**Presentation:** 3 good
**Contribution:** 2 fair

**Summary:**

The authors study alternating mirror descent updates for constrained bilinear zero sum games.  Using alternating updates rather than simultaneous updates, the average regret of the method is $O(K^{-2/3})$ for $K$ iterations in comparison to the known  $O(K^{-1/2})$ regret for simultaneous updates. Unlike simultaneous updates, the iterates do not diverge, but rather maintain a constant energy, like the continuous time dynamics that motivate the approach.

**Questions:**

It seems this work does not handle mirror descent in it's more general form because there is no projection and the constraints need to be handled via the domain of the regularizer through the Legendre condition.  Does this work for natural constraints like polytopes?

Related nitpick(Line 149): Mirror Descent and FTRL are not the same in online learning; these are actually different when constraints are involved that require projection.

**Limitations:**

There are no societal limitations of the work, but limitations of some of the assumptions could probably be tackled more directly.

**Strengths And Weaknesses:**

Strengths
- Bilinear zero sum games are a fundamental primitive in game theory and machine learning so improved understanding is inherently valuable.
- The $O(K^{-2/3}) average regret bound appears to be novel and state of the art for constrained bilinear zero sum games.
- The connections to continuous time dynamics are very useful for gaining intuition on why alternation is useful and the energy analysis is very elegant.
- While the regret notion in this paper is a bit nonstandard, it is well justified in it's connection to the duality gap.

Weaknesses:
-The third order smoothness condition seems potentially reasonable, but does not seem completely standard and maybe should include more discussion. The analysis relies heavily on this.  The example provided for entropy regularization is appreciated here though, but perhaps more examples should be included.

- The domain of the regularizer seems to need to be the constraint set itself, which is not in general necessary for mirror descent.  This avoids having to deal with Bregman projection but weakens the result as the set of constraints that can be handled is much less general.

- I'm not sold on the setting.  While it's interesting having an understanding of this natural update, alternating updates requires coordination between both players, so it's not completely clear why the backward discretization (proximal mirror descent or clairvoyant multiplicative weights) is actually a problem given this gets $O(K^{-1})$ regret.

- The paper is a mirror descent based generalization of [2].  It seems like the main challenge involved in generalizing this is a nonzero Bregman commuter.  This seems like a fairly minor extension with the third order smoothness assumption.

[2] Bailey, James P., Gauthier Gidel, and Georgios Piliouras. "Finite regret and cycles with fixed step-size via alternating gradient descent-ascent." Conference on Learning Theory. PMLR, 2020.

---

> ### Author Response · Authors · 2022-08-02
> **Response for reviewer SuUA**
>
> The authors thank the reviewer for the helpful comments and questions. Please see our itemized responses below:
>
> > The third order smoothness condition seems potentially reasonable, but does not seem completely standard and maybe should include more discussion. The analysis relies heavily on this. The example provided for entropy regularization is appreciated here though, but perhaps more examples should be included.
>
> Thanks for the comment. Currently the third-order smoothness assumption is required to make the Hamiltonian third-order smooth, so that we can bound the deviation in modified Hamiltonian in each step in Theorem 4.4.  We will include more discussion in the revision.
>
> > The domain of the regularizer seems to need to be the constraint set itself, which is not in general necessary for mirror descent. This avoids having to deal with Bregman projection but weakens the result as the set of constraints that can be handled is much less general.
>
> Yes this is true, our analysis currently assumes the regularizer functions are Legendre functions, so we don't need projection for the constraints. Extending our analysis to using Bregman projection is an interesting future direction.
>
> > I'm not sold on the setting. While it's interesting having an understanding of this natural update, alternating updates requires coordination between both players, so it's not completely clear why the backward discretization (proximal mirror descent or clairvoyant multiplicative weights) is actually a problem given this gets $O(K^{-1})$ regret.
>
>
> Alternating method is a natural update rule. So, the fact that it is not understood and that it behaves in a clearly distinct way from all other methodologies, i.e. simultaneous, optimistic, clairvoyant, showcases the importance of the contribution, even if this update is not optimal in terms of regret.
>
> The issue that alternating updates tend to behave closer to their continuous time analogues in normal-form zero-sum games has been pointed out empirically by Hofbauer in [1] more than 25 years ago but this is the first time to our knowledge that formal evidence is provided to make this link precise and formally showcase the superiority to alternating updates. Alternating updates are also used practically in training GANs as well as in CFR+ for computing equilibria for poker. Hence, this setting is interesting from an ML perspective.
>
> [1] Hofbauer, Josef. "Evolutionary dynamics for bimatrix games: A Hamiltonian system?." Journal of mathematical biology 34.5 (1996): 675-688.
>
> > The paper is a mirror descent based generalization of [2]. It seems like the main challenge involved in generalizing this is a nonzero Bregman commuter. This seems like a fairly minor extension with the third order smoothness assumption.
>
> The update rule in Bailey et al. [2] is linear, which allows nice analysis and results.  But to make progress in any form of classical game setting (e.g. normal form, extensive form games, etc), one has to move toward *non-linear* updates, and this is a first important step that we take in that direction. The presence of constraints turns a linear problem to a nonlinear problem. As a consequence, previous analytical tools no longer suffice, and we need to develop novel proof techniques. We also identify the structure of nonlinearity and connections to deeper theoretical tools, such as skew-gradient flow and symplectic structures, which is why the contribution is by no means incremental.
>
> > It seems this work does not handle mirror descent in it's more general form because there is no projection and the constraints need to be handled via the domain of the regularizer through the Legendre condition. Does this work for natural constraints like polytopes?
>
> Polytope with log-barrier regularizer satisfies our assumptions. The mirror descent update requires solving a convex optimization problem which can be done efficiently, e.g. via Newton’s method.
>
> > Related nitpick(Line 149): Mirror Descent and FTRL are not the same in online learning; these are actually different when constraints are involved that require projection.
>
> Thanks for the comment. We will clarify in the revision.

---

> > ### Comment · Reviewer_SuUA · 2022-08-09
> > **Updating Score**
> >
> > I am increasing my score to 6.  While I do still think the Legendre assumption is restrictive, this work studies a very natural update that merits this study.

---

### Official Review · Reviewer_hLz6 · 2022-07-14

**Rating:** 6
**Confidence:** 3
**Soundness:** 3 good
**Presentation:** 4 excellent
**Contribution:** 2 fair

**Summary:**

This paper studies the regret of alternating mirror descent in constrained min-max bilinear two player games. The authors prove that when both players employ this algorithm, they enjoy a regret of $$O(K^{-2/3})$$. This work builds on previous work [1], in which the  authors prove a constant regret for the class of two player zero sum games in the unconstrained setting.

**Questions:**

I would like to ask the authors some clarifying questions.
In lines 150-153 the authors say that the function $\phi$ needs to be continuously differential error, $|| \phi(p)|||\to\infty$ as p approaches the boundary of $P$ and $\nabla \phi$ is a bijection from $P$ to the range $\nabla\phi(P)\subseteq \mathbb{R}^m$.  If I am not mistaken, this means that we cannot $\phi$ to be for example $1/2||p||^2$ which would lead to the projected gradient descent algorithm. Which are the choices that are actually permitted?

Furthermore, in section for you consider that the energy function separable. Even though you claim that this is not necessary and the function H could be anything, it’s not clear to me how the updates would be performed in this case. Building on this, in theorem 4.4 you give an example of a function which is not separable and no example of a function that it is. Which are the energy functions that are actually satisfy the assumptions present in theorem 4.4 and are decomposable?

**Limitations:**

This paper has no negative societal impact to my knowledge.

**Strengths And Weaknesses:**

This paper is in general well written and has a nice flow. The techniques used are quite interesting and extend the ideas presented in [1] to the constrained setting.
However, considering the existence of the results for general-sum multiplayer games  and the result presented in [2], in which the authors prove that when players run Optimistic Dual Average with an adaptive step size (which is easy to deduce that also holds for a step size of the order $1/\sqrt{T}$) then they enjoy constant regret in the class of all convex-concave zero sum games, which includes the bilinear case.

[1]: Bailey, James P., Gauthier Gidel, and Georgios Piliouras. "Finite regret and cycles with fixed step-size via alternating gradient descent-ascent." Conference on Learning Theory. PMLR, 2020.

[2]: Hsieh, Yu-Guan, Kimon Antonakopoulos, and Panayotis Mertikopoulos. "Adaptive learning in continuous games: Optimal regret bounds and convergence to Nash equilibrium." Conference on Learning Theory. PMLR, 2021.

Minor comments:
Line 45: double “the”

---

> ### Author Response · Authors · 2022-08-02
> **Response for reviewer hLz6**
>
> The authors thank the reviewer for the helpful comments and questions. Please see our itemized responses below:
>
> > This paper is in general well written and has a nice flow. The techniques used are quite interesting and extend the ideas presented in [1] to the constrained setting. However, considering the existence of the results for general-sum multiplayer games and the result presented in [2], in which the authors prove that when players run Optimistic Dual Average with an adaptive step size (which is easy to deduce that also holds for a step size of the order $1/\sqrt{T}$) then they enjoy constant regret in the class of all convex-concave zero sum games, which includes the bilinear case.
>
> Thanks for the comment. Our focus is not only to minimize regret, but also to study a natural algorithm, i.e. alternating mirror descent in the context of zero-sum games. Alternating updates have been used successfully for AI applications, e.g. CFR+ in poker, but their formal behavior is still not well understood, see also the discussion section in Bailey et. al. [1].
>
> > Minor comments: Line 45: double “the”
>
> Revised. Thanks!
>
> > I would like to ask the authors some clarifying questions. In lines 150-153 the authors say that the function $\phi$ needs to be continuously differential error, $\|\phi(p)\|\to\infty$ as $p$ approaches the boundary of $P$ and $\nabla \phi$ is a bijection from $P$ to the range $\nabla \phi(P) \subseteq \mathbb{R}^m$. If I am not mistaken, this means that we cannot $\phi$ to be for example $1/2\|p\|^2$ which would lead to the projected gradient descent algorithm. Which are the choices that are actually permitted?
>
> Yes our analysis currently does not cover projected gradient descent. An example of a permitted choice is entropy on the simplex, so mirror descent is the multiplicative weight update algorithm.
>
> > Furthermore, in section for you consider that the energy function separable. Even though you claim that this is not necessary and the function H could be anything, it’s not clear to me how the updates would be performed in this case. Building on this, in theorem 4.4 you give an example of a function which is not separable and no example of a function that it is. Which are the energy functions that are actually satisfy the assumptions present in theorem 4.4 and are decomposable?
>
> Thanks for the comment. Our analysis on regret and connection to game dynamics assumes separable $H(x,y) = f(x)+g(y)$. Some of our analysis (e.g. on skew-gradient flow discretization) holds for general convex $H$. We will revise the exposition to be more precise.
>
> The Example after Theorem 4.4 should read: $H(z) = \log \sum_{i=1}^m e^{x_i} + \log \sum_{j=1}^n e^{y_j}$ for $z = (x,y)$. This is separable, and arises from using the negative entropy regularizer on the simplex: $\phi(p) = -\sum_{i=1}^m p_i \log p_i$ and $\psi(q) = -\sum_{j=1}^n q_j \log q_j$.

---

> > ### Comment · Reviewer_hLz6 · 2022-08-07
> > **Response to authors**
> >
> > Thank you for your response.
> > I maintain my score. I would suggest that the authors clarify which functions can be used in each case.

---

### Official Review · Reviewer_x9Yh · 2022-07-25

**Rating:** 6
**Confidence:** 3
**Soundness:** 3 good
**Presentation:** 3 good
**Contribution:** 3 good

**Summary:**

The paper considers alternating mirror descent of bilinear 2-player zero-sum games in the constrained setting where each player can play within a compact and convex set. It is shown through a suitable reduction to the skew gradient flow dynamics that the average iterates converges to a Nash equilibrium at a speed K^{-2/3} where K is the no. of iterations.



**Questions:**

- In general, you need to explain what is the role of your assumptions formulated in Thm 3.2 (both boundedness and smoothness of third derivative), in order to prove your results, e.g., as compared to simultaneous mirror descent.
No comments whatsoever are provided in the main body of the paper, which is a bit awkward.
- L. 330. Where are you using within Thm 4.4 the assumptions of Thm 3.2 ?

- Regret definition: I'm not sure why regret is defined that way (with two consecutive iterates), I'm not sure I can buy the argument that one player sees the other player's iterate twice. Does it make any difference if we replace (p_k+p_{k+1})/2 in (6) by p_k only [and same for (7)] ?


Minor points:
- L. 45 the the
- L. 235, pls define ||.||_op  (it is only defined in L. 313)
- L. 250: pls check/rephrase this sentence
- L. 303 and 316: Eq (11) does not seem to exist (or, at least, I cannot find it)

**Limitations:**

Limitations are described in the concluding section and in the checklist. I would have liked to see a more thorough discussion of the limitations implied by the assumptions in Thm 3.2.


**Strengths And Weaknesses:**

Strengths:
- the paper is very well written
- the analysis is carried out with care and clarity
- the extension to the constrained case is interesting, yet, see below

Weaknesses:
- The assumptions on boundedness of Bregman divergence(s) are a bit clashing with your assumptions of phi and psi being Legendre. For instance, your result does not seem to apply to the case where the domain P and Q are the probability simplex and the divergence is KL, as KL is unbounded.

---

> ### Author Response · Authors · 2022-08-02
> **Response to reviewer x9Yh**
>
> The authors thank the reviewer for the helpful comments and questions. Please see our itemized responses below:
>
> > The assumptions on boundedness of Bregman divergence(s) are a bit clashing with your assumptions of phi and psi being Legendre. For instance, your result does not seem to apply to the case where the domain P and Q are the probability simplex and the divergence is KL, as KL is unbounded.
>
> Thanks for this comment. The assumption of bounded Bregman divergence is only to bound the final regret independent of the initial point. Without this assumption, the upper bound in Theorem 3.2 can be stated as follows: The regret with respect to a reference point $(p,q)$ has the bound: $R_K(p,q) \le \frac{D_\phi(p,p_0) + D_\psi(q,q_0)}{\eta} + \frac{4 \alpha_{\max}^3 M^4}{3} \eta^2 K$. This is similar to Bailey, Gidel, Piliouras [2] Theorem 2, where the regret is with respect to a fixed static strategy. We will clarify this in the revision.
>
>
>
> > In general, you need to explain what is the role of your assumptions formulated in Thm 3.2 (both boundedness and smoothness of third derivative), in order to prove your results, e.g., as compared to simultaneous mirror descent. No comments whatsoever are provided in the main body of the paper, which is a bit awkward.
>
> The boundedness of Bregman divergence is not necessary (kindly see above). The boundedness of domain is to make the Hamiltonian Lipschitz; third-order smoothness of $\phi^*$ is to make the Hamiltonian third-order smooth. Both conditions are needed in Theorem 4.4 to bound the deviation in modified Hamiltonian in each step.
>
>
>
> > L. 330. Where are you using within Thm 4.4 the assumptions of Thm 3.2 ?
>
> Theorem 4.4 assumes H to be Lipschitz and third-order smooth. In Theorem 3.2, $H(x,y) = \phi^*(x) + \psi^*(y)$. The gradient of $H$ is the iterate in the original domain: $\nabla H(x,y) = (p,q)$, so bounded domain implies $H$ is Lipschitz. The third-order smoothness assumption on $\phi^*, \psi^*$ implies $H$ is also third-order smooth.
>
>
>
> > Regret definition: I'm not sure why regret is defined that way (with two consecutive iterates), I'm not sure I can buy the argument that one player sees the other player's iterate twice. Does it make any difference if we replace (p_k+p_{k+1})/2 in (6) by p_k only [and same for (7)] ?
>
> The way the regret is defined here is the appropriate definition for alternative minimization. One alternative way of seeing is multiplying equation (6), ie. the definition of regret, by a factor of two: $\sum_{k=0}^{K-1} (p_k+p_{k+1})^\top A q_k - \sum_{k=0}^{K-1} 2p^\top A q_k$. Now the first term expresses the total payoff enjoyed by the player over the whole time history. The second expresses that players aggregate payoff if they play a fixed strategy $p$. This is the standard definition of regret in online setting. The reason the term 2 appears is because their opponent always repeats every strategy twice, since there is a setting of alternating updates. See also the similar definition in the paper by Bailey, Gidel and Piliouras [2].
>
>
>
> > Minor points ...
>
> Thanks for the comments. We will clarify in the revision.

---

### Meta-Review · Area_Chair_U9QF · 2022-09-07

**Recommendation:** Accept
**Confidence:** Less certain

**Metareview:**

The paper studies the regret of alternating mirror descent in constrained bilinear 2-player zero-sum games where each player can play within a compact and convex set. It is shown through a suitable reduction to the skew gradient flow dynamics that the average iterates converges to a Nash equilibrium at a speed K^{-2/3} where K is the no. of iterations.This work builds on previous work [1], in which the authors prove a constant regret for the class of two player zero sum games in the unconstrained setting.
The reviewers agreed during the post-discussion that this is a non-trivial paper that extends known results to the constrained setting in some meaningful way (yet, paying the price of some extra assumptions and limitations on the algorithms played by the two players).
To the authors: Please follow the reviewers' suggestions to improve presentation (like, more context and more discussion about the limitations of these results).

**Award:**

No

---

### Decision · Program_Chairs · 2022-09-14

Accept